# Housing-temperature reveals energy intake counter-balances energy expenditure in normal-weight, but not diet-induced obese, male mice

Linu Mary John[1], Natalia Petersen[1], Marina Kjærgaard Gerstenberg[1], Lola Torz [1,2], Kent Pedersen[1], Berit Østergaard Christoffersen[1] & Rune Ehrenreich Kuhre [1,3 ✉]

Most metabolic studies on mice are performed at room temperature, although under these conditions mice, unlike humans, spend considerable energy to maintain core temperature. Here, we characterize the impact of housing temperature on energy expenditure (EE), energy homeostasis and plasma concentrations of appetite- and glucoregulatory hormones in normal-weight and diet-induced obese (DIO) C57BL/6J mice fed chow or 45% high-fat-diet, respectively. Mice were housed for 33 days at 22, 25, 27.5, and 30 °C in an indirect-calorimetry-system. We show that energy expenditure increases linearly from 30 °C towards 22 °C and is ~30% higher at 22 °C in both mouse models. In normal-weight mice, food intake counter-balances EE. In contrast, DIO mice do not reduce food intake when EE is lowered. By end of study, mice at 30 °C, therefore, had higher body weight, fat mass and plasma glycerol and triglycerides than mice at 22 °C. Dysregulated counterbalancing in DIO mice may result from increased pleasure-based eating.

[1] Global Obesity and Liver Disease, Global Drug Discovery, Novo Nordisk A/S, Måløv, Denmark. [2] Department of Veterinary and Animal Science, Faculty of Health and Medical Sciences, University of Copenhagen, DK-2200 Copenhagen, Denmark. [3] Department of Biomedical Sciences, Faculty of Health and Medical Sciences, University of Copenhagen, DK-2200, Copenhagen, Denmark. ✉email: RUKU@novonordisk.com

Mice are the most frequently used animal model to investigate human physiology and pathophysiology, and are often the default animal used in early drug discovery and drug development[1]. Yet, mice differ from humans on several important physiological parameters and while allometric scaling can be used to some degree for translation to humans, the considerable disparity in size between mice and humans presents a fundamental incongruity regarding body temperature regulation and energy homeostasis. The average weight of a mature mouse is at least a thousand times smaller than an adult human (50 g vs. 50 kg), whereas the surface area: weight ratio differs by ~400 fold due to the nonlinear geometric transformation described by Meeh's formula[2]. As a result, a mouse loses significantly more heat relative to its volume, and is consequently more temperature sensitive and prone to hypothermia with an average basal metabolic rate ten times that of humans[3]. At standard room temperature (~22 °C) housing conditions, a mouse must increase total energy expenditure (EE) by ~30% to maintain core body temperature. At lower temperatures, EE is increased even further with an increase of ~50 and 100% at 15 and 7 °C relative to EE at 22 °C[4]. Standard housing conditions therefore induces cold-stress responses, and this may jeopardize the translatability of experimental results generated in mice to humans, since humans living in modernized societies live most of our lives at thermoneutrality (because our lower surface area: volume ratio makes us less temperature sensitive and because we create thermo-neutral zones (TNZ) around us)[3,5]. Therefore, the thermoneutral zone in humans (defined as the temperature interval where a healthy and undressed human adult can maintain normal body temperature without needing to increase EE above basic metabolic rate) spans from ~19 to 30 °C[6] whereas the minimum temperature of thermoneutrality in mice is higher and the interval more narrow, only spanning 2–4 °C[7,8]. Indeed, this important aspect has recently received considerable attention[4,7–12], and it has been suggested that some of the 'species differences' may be alleviated by increasing the housing temperature[9]. However, there exists no consensus on the temperature range constituting thermoneutrality in mice. Thus, it remains debated whether the lower critical temperature within the thermoneutral range for single housed mice is closer to 25 °C or to 30 °C[4,7,8,10,12]. Most of the studies investigating the effect of ambient temperature on EE and other metabolic parameters are restricted in time span from hours to a few days, and it is thus less clear to what extent prolonged housing at different temperatures may affect metabolic parameters such as body weight, food intake, substrate utilization, glucose tolerance as well plasma concentrations of lipids and gluco- and appetite-regulating hormones. Moreover, it warrants further investigation to what extent these parameters may be influenced by diet (with DIO mice on high-fat diet potentially being more driven by pleasure-based (hedonic) eating). To provide more information on this subject, we investigated the importance of housing temperature for the abovementioned metabolic parameters in adult normal-weight male on chow and diet-induced obese (DIO) male mice on 45% high-fat diet. Mice were housed for at least for three weeks at either 22, 25, 27.5, or 30 °C. Temperatures below 22 °C were not studied as housing temperatures in standard animal facilities are rarely below room temperature. We find that single-housed normal-weight and DIO mice respond similarly to changes in housing temperature in terms of EE, and that the lower point of thermoneutrality is above 27.5 °C in both mouse models regardless of housing conditions (with or without hide/nesting material). However, while normal-weight mice on chow adjust food intake in accordance with EE, food intake in DIO mice are largely independent of EE, causing mice housed at 30 °C to gain more weight than mice at 22 °C. Consistent with body weight

data, plasma concentrations of lipids and ketone bodies indicated that DIO mice at 30 °C were in a more positive energy balance than mice at 22 °C. The underlying reasons for the differences in counter-balancing energy intake with EE between normal-weight and DIO mice requires further investigation but is presumably a result of pathophysiological changes in the DIO mice combined with higher pleasure-based eating due to the obesogenic diet.

## Results

**Temporal ambient temperature changes within animal, normal-weight mice**. EE increased linearly from 30 to 22 °C and was ~30% higher at 22 °C compared to 30 °C (Fig. 1a, b). Respiratory exchange ratio (RER) was not affected by temperature (Fig. 1c, d). Food intake matched the dynamics in EE and increased with decreasing temperature (and was also approximately 30% higher at 22 °C compared to 30 °C (Fig. 1e, f). Water intake and activity level were not consistently affected by temperature (Fig. 1g–j).

**Temporal ambient temperature changes within animal, DIO mice**. As in the case of normal-weight mice, EE increased in a linear fashion with decreasing temperature and were in this case also ~30% higher at 22 °C compared to 30 °C (Fig. 2a, b). RER did not vary at different temperatures (Fig. 2c, d). Unlike the case with normal-weight mice, food intake was not consistently matched with EE across housing temperatures. Food intake, water intake, and activity level were not consistently influenced by temperature (Fig. 2e–j).

**Constant ambient temperatures, normal-weight mice**. In another line of experiments, we examined the importance of ambient temperature for the same parameters but on this occasion the comparison was made across groups of mice housed constantly at specified temperatures. Mice were distributed into four groups such that statistical variations in the mean and standard deviations of body weight, fat and normal-weight mass were minimized (Fig. 3a–c). After seven days of acclimation, EE was recorded for 4.5 days. EE was markedly affected by ambient temperature in both light and dark phase (Fig. 3d) and increased in a linear manner with decreasing temperature from 27.5 °C towards 22 °C (Fig. 3e). RER was slightly reduced in the 25 °C group compared to the other groups with no difference between the remaining groups (Fig. 3f, g). Food intake paralleled the patterns in EE a was increased by ~30% at 22 °C compared to 30 °C (Fig. 3h, i). Water intake and activity level was not significantly different between groups (Fig. 3j, k). Housing at different temperatures up to 33 days did not lead to group differences in body weight, fat-free mass, and fat mass (Fig. 3n–s), but compared to own stating points, fat-free weight mass was reduced approximately 15% (Fig. 3b, p, q)) and fat mass more than doubled (from ~1 g to 2–3 g, Fig. 3c, r, s). Unfortunately, the 30 °C cabinet had a miscalibration precluding presentation of accurate EE and RER data.

**Constant ambient temperatures, DIO mice**. At the start of the study, mice were matched on body weight, fat free mass, and fat mass Fig. 4a–c) and were, like the study in normal-weight mice, housed at 22, 25, 27.5, and 30 °C. When comparing mice across groups, the relationship between EE and temperature showed a similar linear relationship as when the temperature was changed temporally in the same mice. Mice housed at 22 °C thus spend ~30% more energy than mice housed at 30 °C (Fig. 4d, e). As when studying the within animal effects, RER was not consistently influenced by temperature (Fig. 4f, g). Food intake, water intake, and activity were not significantly influenced by temperature (Fig. 4h–m). After 33 days of housing, mice at 30 °C had

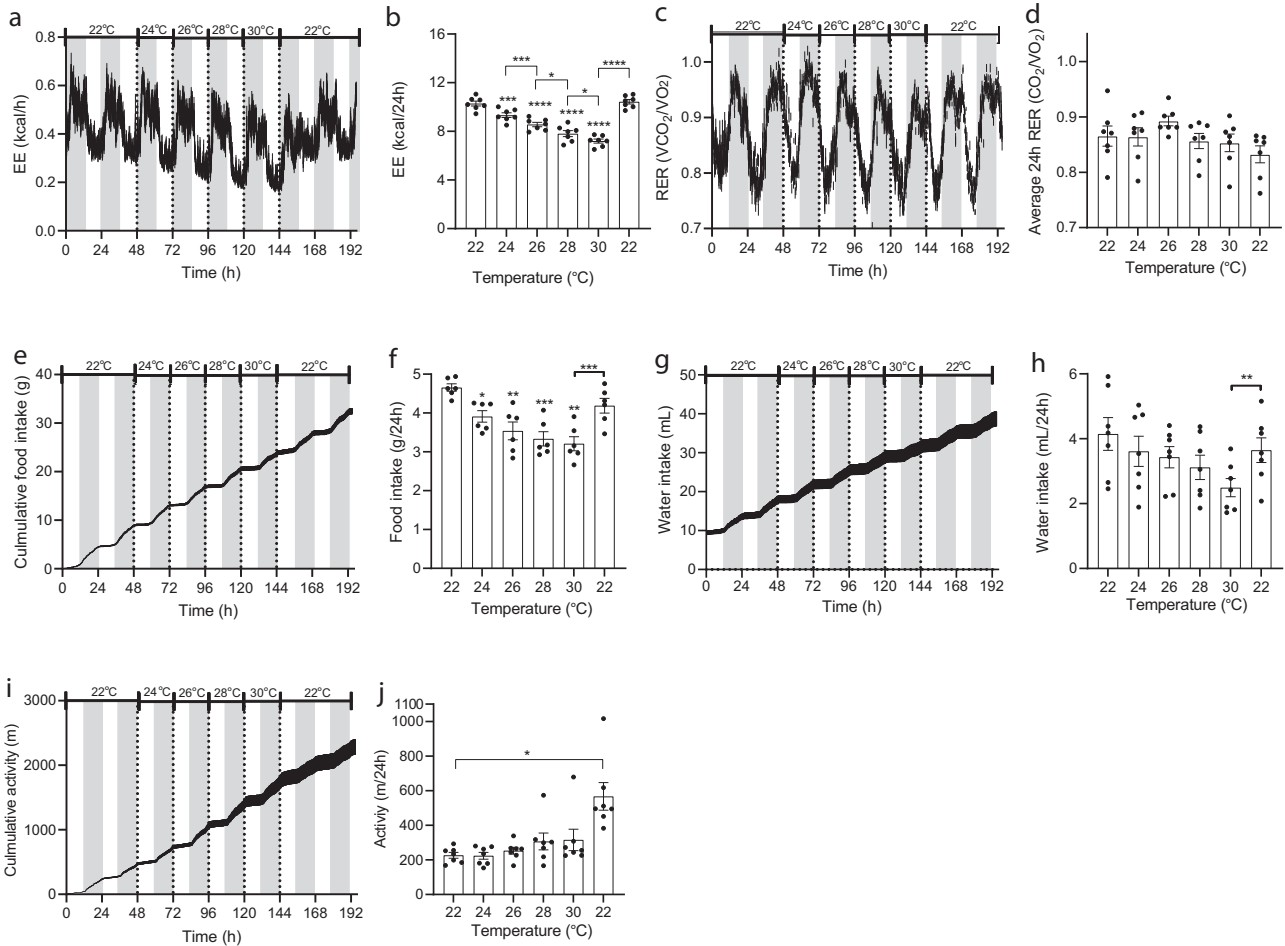

**Fig. 1 Effect of housing temperature on energy expenditure, respiratory exchange ratio, food intake, water intake, and activity level in adult normal-weight male mice.** Male mice (C57BL/6J, 20 weeks, individually housed, $n = 7$) were housed in metabolic cages at 22 °C for a week preceding study initiation. After two days of collection of baseline data, temperature was increased in 2 °C increments each day at 0600 h (start of light phase). Data are presented as means ± SEM and dark phase (1800-0600h) is indicated with grey boxes. **a** Energy expenditure (kcal/h), **b** total energy expenditures at different temperatures (kcal/24 h), **c** Respiratory exchange ratios ($VCO_2/VO_2$: scale 0.7–1.0), **d** average RER ($VCO_2/VO_2$) during light and dark phase (zero-value defined as 0.7). **e** Cumulative food intakes (g), **f** total food intakes during 24 h, **g** cumulative water intake (mL), **h** Total water intakes during 24 h, **i** cumulative activity level (m), and **j** total activity levels (m/24 h). Mice were at the indicated temperatures for 48 h. Data shown for 24, 26, 28, and 30 °C are from the last 24 h of each period. Mice remained of chow throughout the study. Statistical significance was tested by One-way ANOVA for repeated measurements followed by Tukey multiple comparison test. Stars indicate significance from initial 22 °C values, hatches indicate significance between other groups as indicated. *$P < 0.05$, **$P < 0.01$, **$P < 0.001$, ****$P < 0.0001$. Averages were calculated over the entire experimental period (0–192 h). $n = 7$.

a significantly higher body weight than mice housed at 22 °C (Fig. 4n). Compared to their respective starting points, mice housed at 30 °C had thus gained significantly more weight than mice housed at 22 °C (means ± SEM: Fig. 4o). The relative higher weight gain was driven by increased fat mass (Fig. 4p, q), but not by increased fat-free mass (Fig. 4r, s). Consistent with the lower EE at 30 °C, BAT expression of some genes that increases BAT function/activity was reduced at 30 °C compared to 22 °C: *Adra1a, Adrb3,* and *Prdm16*. Other key genes that are also increases BAT function/activity was unaffected: *Sema3a (neurite growth regulation), Tfam (mitochondrial biogenesis), Adrb1, Adra2a, Pck1 (gluconeogenesis),* and *Cpt1a*. Surprisingly, *Ucp1* and *Vegf-a*, associated with increased thermogenic activity[13,14], were not reduced in the 30 °C group. In fact, three mice showed higher levels of *Ucp1* than mice in 22 °C group and *Vegf-a* and *Adrb2* were significantly upregulated. In mice housed at 25 and 27.5 °C changes compared to the 22 °C group was not detected (Supplementary Fig. 1).

**Importance of house and bedding material for EE and RER in normal-weight mice.** Like humans, mice often create microenvironments to decrease heat loss to the surroundings. To quantitatively investigate the importance of such environments for EE, we evaluated EE at 22, 25, 27.5, and 30 °C in the absence or presence of hide enclosures and nesting material. At 22 °C, addition of a standard hides decreased EE by ~4%. Subsequent addition of nesting material reduced EE by another 3–4% Fig. 5a, b). No significant changes in RER, food intake, water intake, or activity level were observed with addition of house or hide +bedding material (Fig. 5i–p). At 25 and 30 °C, adding a hide and nesting material also significantly reduced EE but responses were quantitatively smaller. At 27.5 °C, no differences were observed. Notably, EE decreased in these experiments more with increasing temperatures and was in this case ~57% lower at 30 °C compared to 22 °C (Fig. 5c–h). Conducting the same analysis on the light phase only, where EE is closer to basal metabolic rate, since the mice in this case mostly rest inside their hide, resulted in

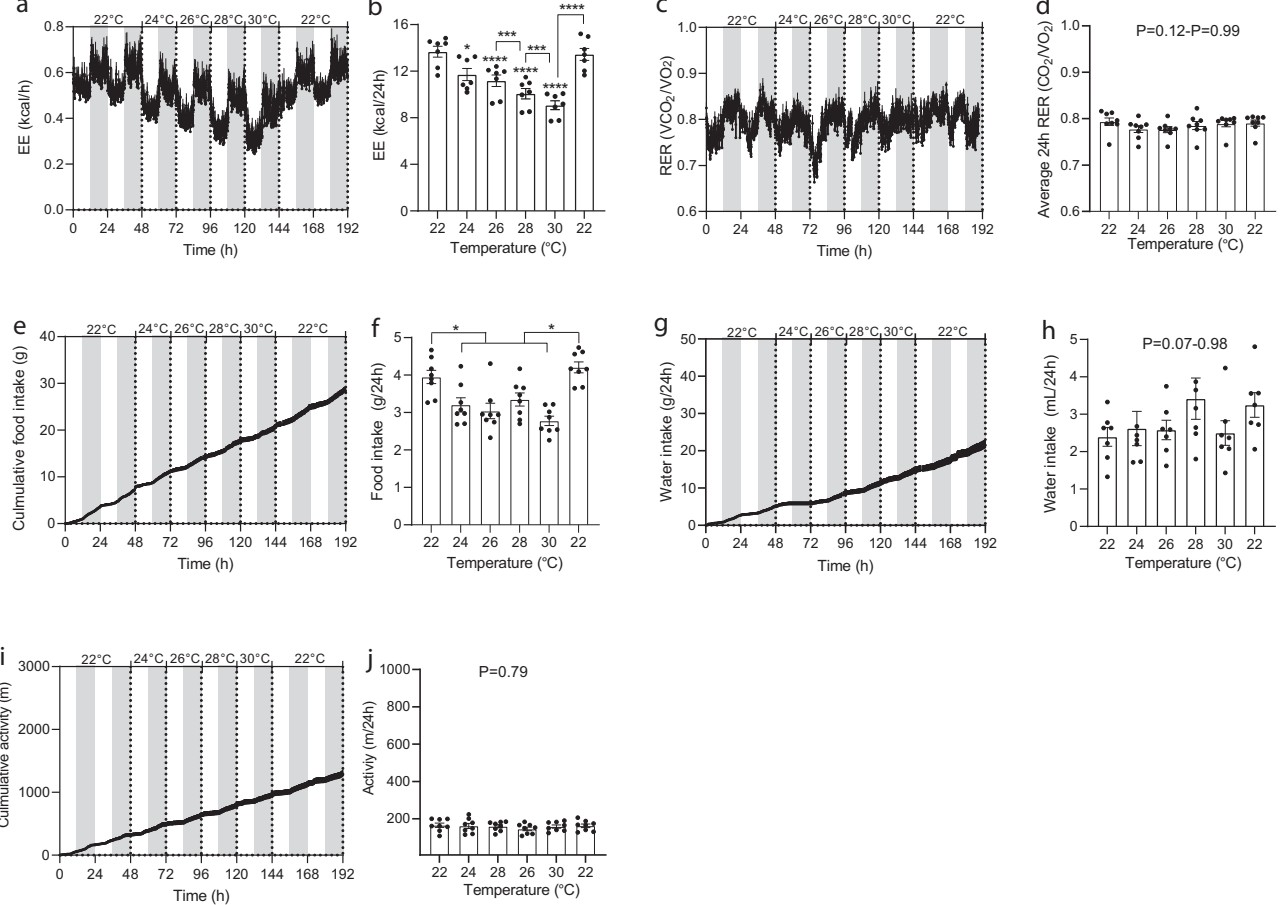

**Fig. 2 Effect of housing temperature on energy expenditure, respiratory exchange ratio, food intake, water intake, and activity level in adult normal-weight male mice and DIO male mice.** Male (C57BL/6J, 20 weeks) DIO mice were housed individually in metabolic cages at 22 °C for a week preceding study initiation. Mice had ad libitum access to 45% HFD. After acclimatization, baseline data was collected for two days. Hereafter, temperature was increased in 2 °C increments every other day at 0600 h (start of light phase). Data are presented as means ± SEM and dark phase (1800-0600h) is indicated with grey boxes. **a** Energy expenditure (kcal/h), **b** total energy expenditures at different temperatures (kcal/24 h), **c** respiratory exchange ratios ($VCO_2/VO_2$: scale 0.7-1.0), **d** average RER ($VCO_2/VO_2$) during light and dark phase (zero-value defined as 0.7). **e** Cumulative food intakes (g), **f** total food intakes during 24 h, **g** cumulative water intake (mL), **h** total water intakes during 24 h, **i** cumulative activity level (m), and **j** total activity levels (m/24 h). Mice were at the indicated temperatures for 48 h. Data shown for 24, 26, 28, and 30 °C are from the last 24 h of each period. Mice remained on 45% HFD until end of study. Statistical significance was tested by One-way ANOVA for repeated measurements followed by Tukey multiple comparison test. Stars indicate significance from initial 22 °C values, hatches indicate significance between other groups as indicated. *$P < 0.05$, ***$P < 0.001$, ****$P < 0.0001$. Averages were calculated over the entire experimental period (0–192 h). $n = 7$.

comparable effect sizes across temperatures (Supplementary Fig. 2a–h).

**Effect of ambient temperatures on plasma lipids, leptin, insulin, FGF21, and glucagon during fasting and on post OGTT glycaemic control.** In normal-weight mice (fasted 2–3 h), housing at the different temperatures did not result in significant differences in plasma concentrations of TG, 3-HB, cholesterol, ALT and AST, but HDL varied in a temperature-independent manner (Fig. 6a–f). Fasting plasma concentrations of leptin, insulin, C-peptide, and glucagon was also not different between groups (Fig. 6g–j). At the day of glucose tolerance challenge (after 31 days of housing at the different temperatures), blood glucose at baseline (5–6 h fasting) was ~6.5 mM and did not differ between groups. Administration of oral glucose increased blood glucose concentrations significantly in all groups, but both peak concentration and incremental area under the curves (iAUCs) (15–120 min) were lower in the group of mice housed at 30 °C (individual time points: $P < 0.05$–$P < 0.0001$, Fig. 6k, l) compared

to the mice housed at 22, 25 and 27.5 °C (which did not differ amongst each other).

*In DIO mice* (also fasted 2–3 h), plasma concentrations of cholesterol, HDL, ALT, AST, and FFA did not vary between groups. TG and glycerol were both significantly higher in the 30 °C group compared to the 22 °C group (Fig. 7a–h). In contrast, 3-HB was ~25% lower in the 30 °C compared to the 22 °C (Fig. 7b). Thus, although mice housed at 22 °C had overall been in a positive energy balance inferred based on weight gain, the differences in plasma concentrations of TG, glycerol, and 3-HB indicated that mice at 22 °C were, at the time of sample collection, in a relative more energy negative state than mice housed at 30 °C. Consistent with this, liver concentrations of extractable glycerol and TG, but not glycogen and cholesterol, were higher in the 30 °C group (Supplementary Fig. 3a–d). To investigate whether the temperature-dependent differences in lipolysis (judged by plasma TG and glycerol) was a result of intrinsic alterations in epididymal or inguinal fat, we extracted fat tissue by end of the study from these stores and quantified FFA and glycerol release ex vivo. In all experimental groups, adipose tissue

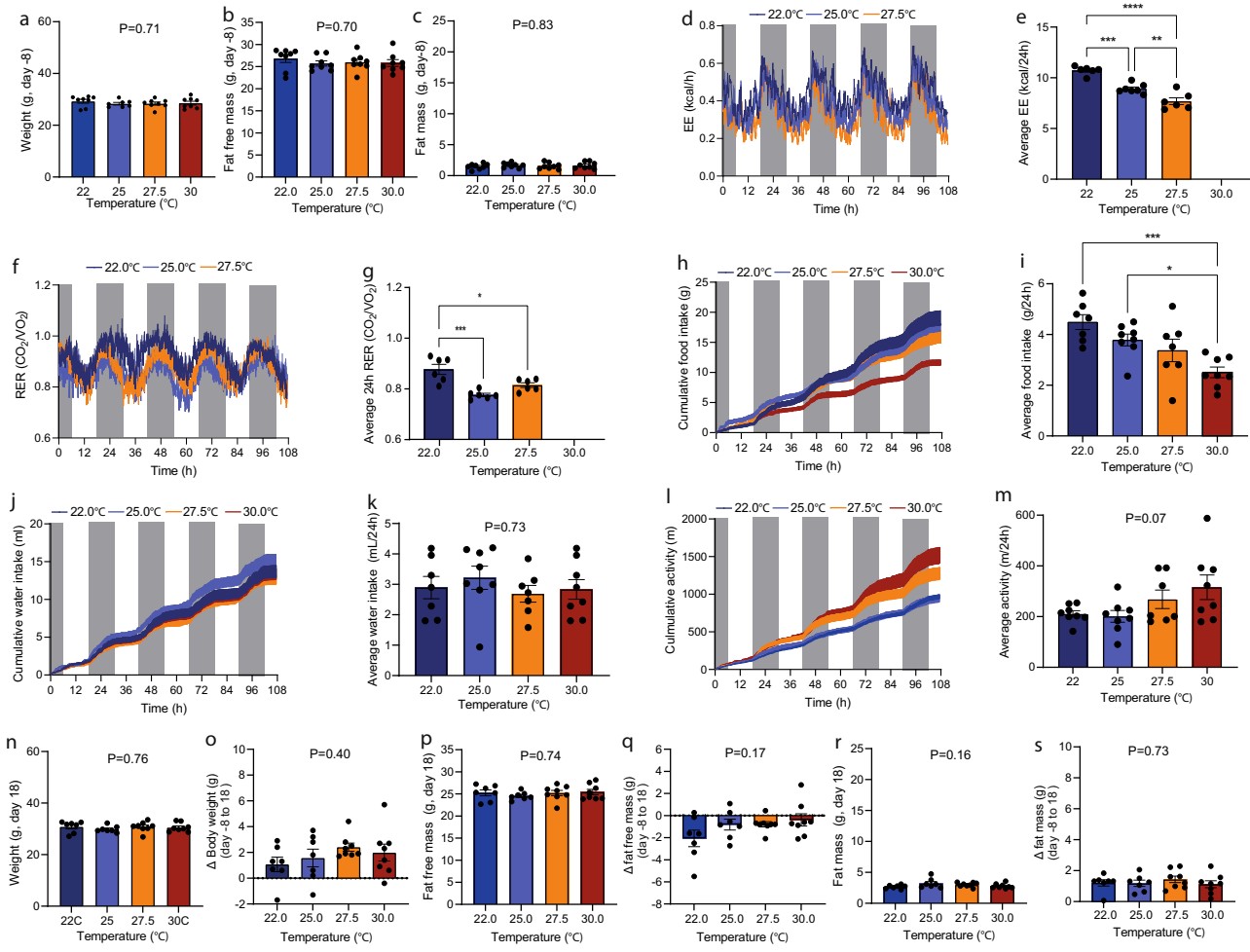

**Fig. 3 Expenditure, respiratory exchange ratio, food intake, water intake, and activity level in adult normal-weight male mice matched on age, weight, fat, and normal-weight mass.** Weight (**a**), Fat free mass (**b**), and fat mass (**c**) at -8 day (the day before transfer to SABLE system). **d** Energy expenditure (kcal/h). **e** Average energy expenditures (0–108 h) at different temperatures (kcal/24 h). **f** Respiratory exchange ratios (RER) ($VCO_2/VO_2$). **g** Average RER ($VCO_2/VO_2$). **h** Cumulative food intakes (g). **i** Average food intakes (g/24 h). **j** Cumulative water intake (mL). **k** Average water intake (mL/24 h). **l** Cumulative activity level (m). **m** Average activity level (m/24 h). **n** Weight at day 18, **o** changes in weight (from day -8 to 18), **p** fat free mass at day 18, **q** changes in fat free mass (from day -8 to 18), **r** fat mass at day 18, and **s** changes in fat mass (from day -8 to day 18). Statistical significance was tested by Oneway-ANOVA for repeated measurements followed by Tukey multiple comparison test. *$P < 0.05$, **$P < 0.01$, ***$P < 0.001$, ****$P < 0.0001$. Data are presented as means + SEM and dark phase (1800-0600h) is indicated with grey boxes. Dots in bar graphs indicate individual mice. Averages were calculated over the entire experimental period (0–108 h). $n = 7$.

samples from epididymal and inguinal depots showed increased glycerol and FFA production in response to isoproterenol stimulation by at least two-fold (Supplementary Fig. 4a–d). However, no effect of housing temperatures was detected for basal or for isoproterenol-stimulated lipolysis. Plasma leptin was, consistent with the higher weight and fat mass, significantly higher in the 30 °C group compared to the 22 °C group (Fig. 7i). On the contrary, plasma insulin and C-peptide did not differ between temperature groups (Fig. 7j, k), but plasma glucagon showed temperature dependency, but in this case in the opposite direction being almost twice as high in the 22 °C group compared to the 30 °C group (Fig. 7l). FGF21 was not different between temperature groups (Fig. 7m). At the day of the OGTT, blood glucose was ~10 mM at baseline and did not differ between mice housed at different temperatures (Fig. 7n). Oral glucose administration increased blood glucose and peaked in all groups at a concentration of ~18 mM, 15 min after administration. Neither iAUCs (15–120 min) nor concentrations at individual time points after administration (15, 30, 60, 90, and 120 min) were significant differences between groups (Fig. 7n, o).

## Discussion

Translatability of rodent data to humans is an intractable challenge that is at the heart of interpreting the significance of observations made in physiological and pharmacological research settings. For economic reasons and ease of conducting studies, mice are generally housed at room temperature, which is below their thermoneutral zone, leading to activation of a variety of compensatory physiological systems that increases metabolic rate and may compromise translatability[9]. Cold exposure in mice, may thus render mice resistant to diet induced obesity[15], and may prevent the manifestation of hyperglycaemia in streptozotocin treated rats due to elevated insulin-independent glucose disposal[16]. It is, however, less clear to what extent prolonged housing at different relevant temperatures, from room temperature to thermoneutrality, affects different energy homeostasis and metabolic parameters in normal-weight mice (on chow) and DIO mice (on HFD) and to what extent they are capable of counter-balancing increased EE with increased food intake. The studies presented herein aimed to provide some clarity on this subject.

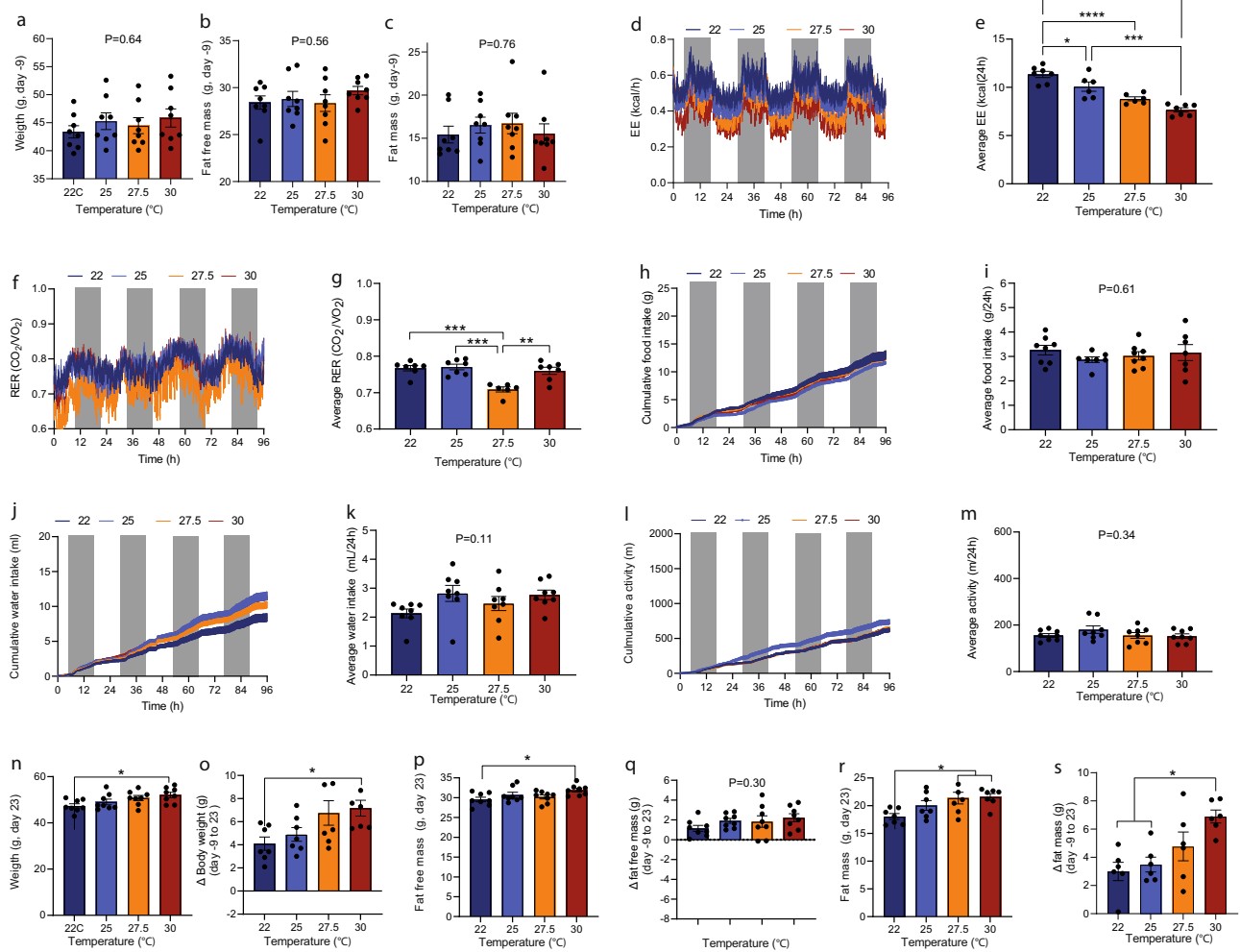

**Fig. 4 Effect of housing temperature on energy expenditure, respiratory exchange ratio, food intake, water intake, and activity level in mid-old DIO male mice.** Weight (**a**), fat free mass (**b**), and fat mass (**c**) at -9 day (the day before transfer to SABLE system). **d** Energy expenditure (EE, kcal/h). **e** Average energy expenditures (0–96 h) at different temperatures (kcal/24 h). **f** Respiratory exchange ratios (RER, $VCO_2/VO_2$). **g** Average RER ($VCO_2/VO_2$). **h** Cumulative food intake (g). **i** Average food intake (g/24 h). **j** Cumulative water intake (mL). **k** Average water intake (mL/24 h). **l** Cumulative activity level (m). **m** Average activity level (m/24 h). **n** weight (g) at day 23, **o** change in body weight; day 23 compared to day -9, **p** fat free mass at day 23, **q** change in fat free mass (g); day 23 compared to day -8, **r** fat mass (g) at day 23, **s** change in fat mass (g), day 23 compared to day -8. Statistical significance was tested by Oneway-ANOVA for repeated measurements followed by Tukey multiple comparison test. *$P < 0.05$, ***$P < 0.001$, ****$P < 0.0001$. Data are presented as means + SEM and dark phase (1800-0600h) is indicated with grey boxes. Dots in bar graphs indicate individual mice. Averages were calculated over the entire experimental period (0–96 h). $n = 7$.

We show that in both adult normal-weight and DIO male mice, EE is inversely and linearly associated with housing temperature in the interval between 22 and 30 °C. Thus, EE was ~30% higher at 22 °C than at 30 °C in both mouse models. However, an important difference between the normal-weight mice and DIO mice was that whereas normal-weight mice matched EE at lower temperatures by adjusting food intake accordingly, food intake in DIO mice was similar across studied temperatures. After a month, DIO mice housed at 30 °C had thus gained more body weight and fat mass than mice housed at 22 °C, whereas housing at the same temperatures and for the same duration did not result in temperature-dependent weight differences for normal-weight mice. The finding that housing at room temperature causes DIO mice, or normal-weight mice on high-fat diet, but not normal-weight mice on chow, to gain a relatively lower amount of weight compared to temperatures close to or at thermoneutrality, is also supported by other studies[17–21], but not by all[22,23].

The ability to create microenvironments to reduce heat loss has been suggested to left-shift the point for thermoneutrality[8,12]. In

our study, the addition of nesting material and hide both reduced EE, but not to an extent causing thermoneutrality to be reached before 28 °C. Therefore, our data do not support that the lower point for thermoneutrality in single housed adult mice, either house with or without environmental enrichments, should be 26–28 °C, as demonstrated[8,12], but it supports other studies suggesting that the lower point of thermoneutrality in the mouse is 30 °C[7,10,24]. To complicate the matter further, it has been shown that the thermoneutral point in the mouse is not static over the course of the day as it is lower during the resting (light) phase, presumably due to lower heat generation by activity- and diet-induced thermogenesis. During light phase, the lower point of thermoneutrality was thus found to be ~29 °C whereas it was ~33 °C during the dark phase[25].

Ultimately, the relationship between ambient temperature and total energy expenditure is driven by heat dissipation. In this context, surface area:volume ratio is an important determining factor for heat sensitivity, influencing heat dissipation (surface area) and heat generation (volume). Heat dissipation is in addition

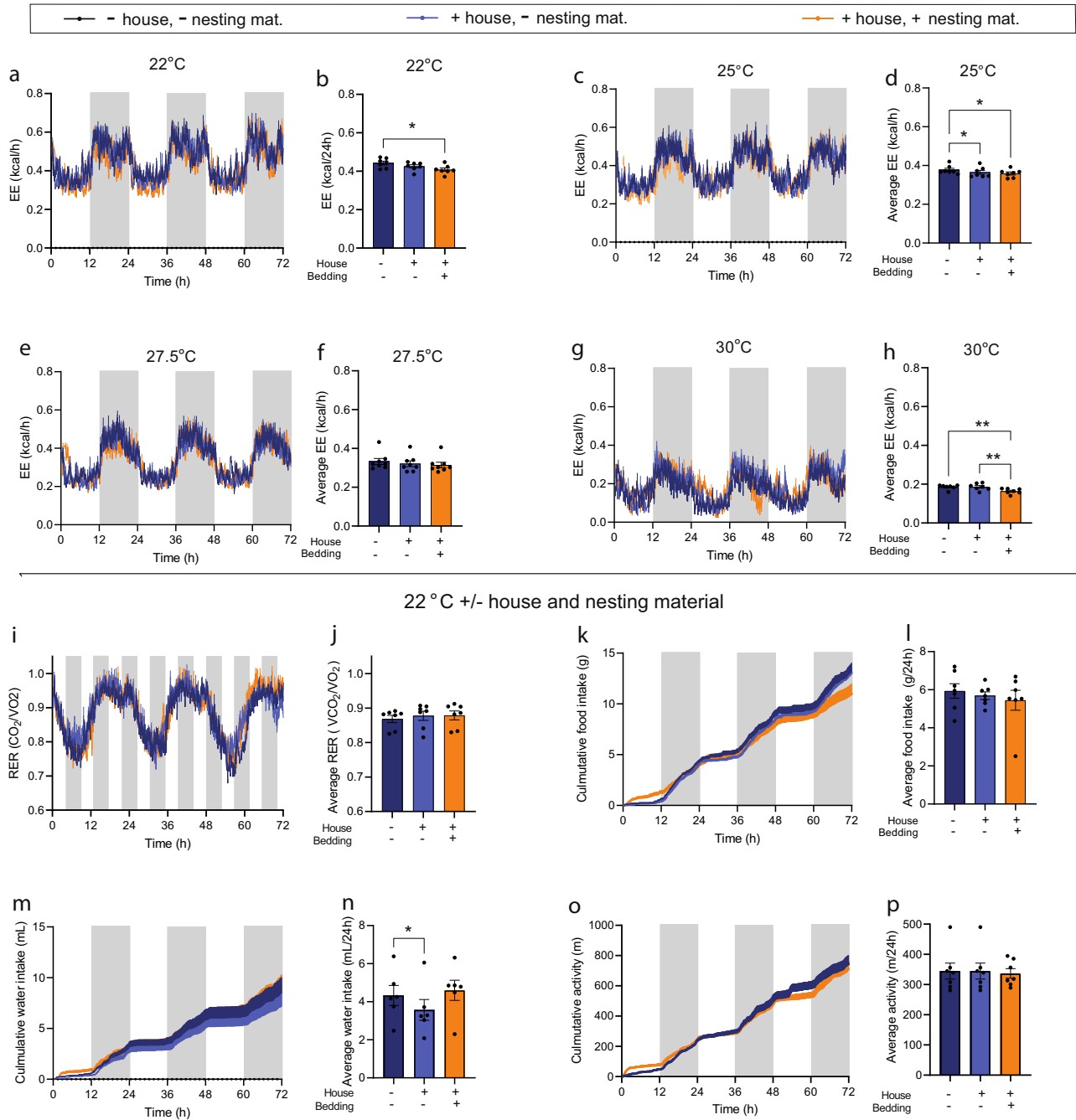

**Fig. 5 Importance of houses and nesting material for expenditure, respiratory exchange ratio, food intake, water intake, and activity level in mid-old normal-weight and DIO male mice.** Data from mice housed hide and nesting material (dark blue), with house but without nesting material (light blue), and with house and with nesting material (orange). **a**, **c**, **e**, and **g** Energy expenditure (EE,kcal/h) at housing at 22, 25, 27.5, and 30 °C, **b**, **d**, **f**, and **h** average EE (kcal/h). **i**–**p** Data from mice housed at 22 °C: **i** Respiratory exchange ratio (RER, $VCO_2/VO_2$), **j** Average RER($VCO_2/VO_2$), **k** cumulative food intake (g), **l** average food intake (g/24 h), **m** Cumulative water intake (mL), **n** average water intake AUCs (mL/24 h), **o** cumulative activity (m), **p** average activity level (m/24 h). Data are presented as means + SEM and dark phase (1800-0600h) is indicated with grey boxes. Dots in bar graphs indicate individual mice. Statistical significance was tested by Oneway-ANOVA for repeated measurements followed by Tukey multiple comparison test. *$P < 0.05$, **$P < 0.01$. Averages were calculated over the entire experimental period (0–72 h). $n = 7$.

to surface area also determined by insulation (rate of heat transfer). In humans, fat mass may reduce heat loss by providing an insulating barrier around the shell of the body[26], and it has been suggested that fat mass may also be important for insulation in mice, lowering the point for thermoneutrality and decreasing the temperature sensitivity when below the thermoneutral point (slope of ambient temperature and EE relationship)[12]. Our study was not designed to directly assess such putative relationship as data on

body composition was collected nine days prior to collection of energy expenditure data, and because fat mass was not stable throughout the study. Nevertheless, since EE was ~30% lower at 30 °C compared to at 22 °C in both normal-weight mice and DIO mice, despite at least a five-fold difference in fat mass, our data do not support that adiposity should provide a major insulating factor, at least not within the temperature range studied. This agrees with other studies that were better designed to investigate this[4,24]. In

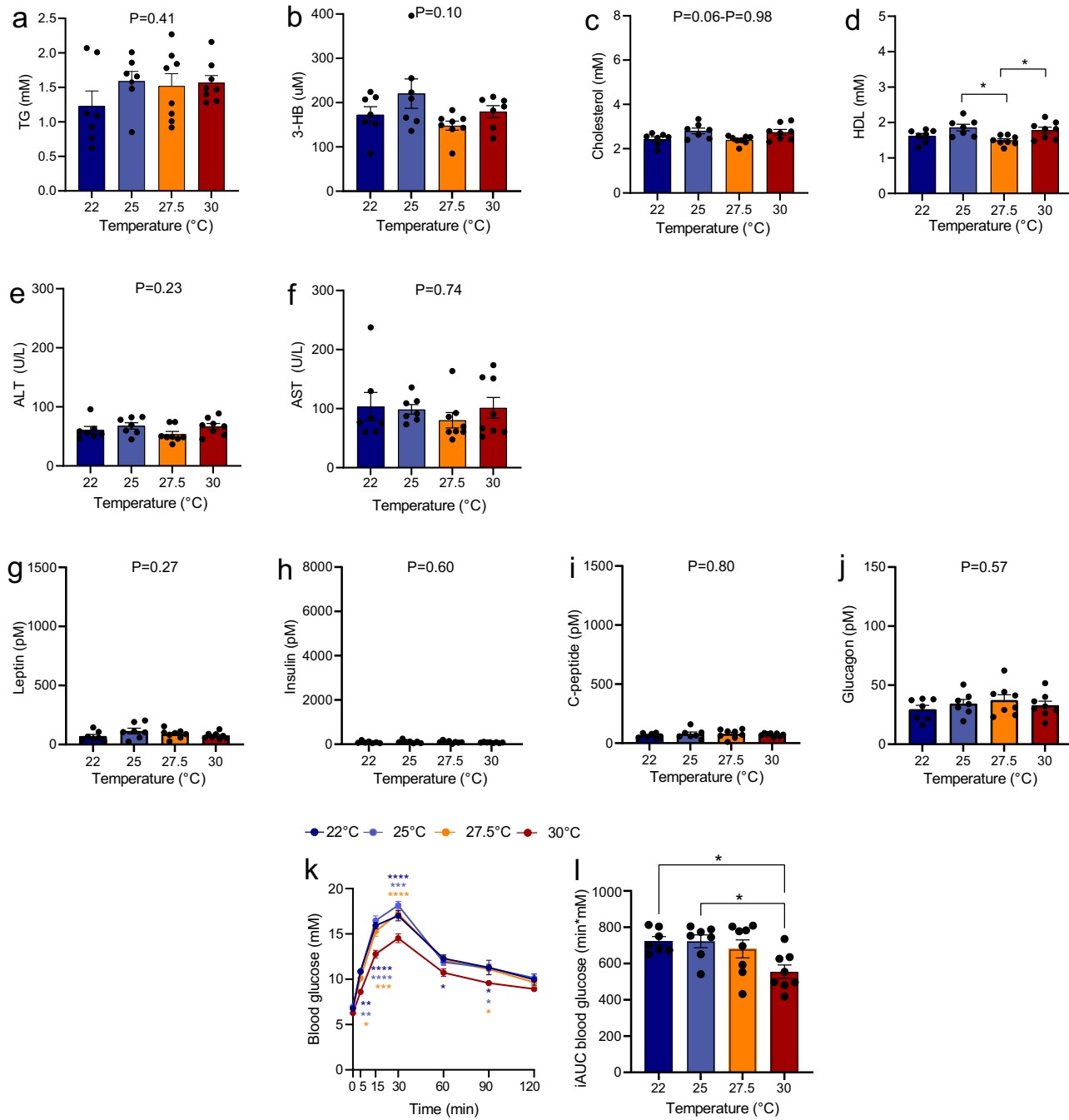

**Fig. 6 Effect of housing temperature on fasting plasma concentrations of lipids, ketone bodies, hepatic inflammatory markers, gluco- and appetite-regulating hormones, and glucose tolerance after an oral glucose in normal-weight mice.** Plasma concentrations of TG, 3-HB, cholesterol, HDL, ALT, AST, FFA, glycerol, leptin, insulin, C-peptide, and glucagon are shown in adult DIO male mice (**a**–**l**) after 33 days of housing at the indicated temperatures. Mice had been fasted 2–3 h prior to collection of blood. The oral glucose tolerance test is an exception to this as it was performed two days before study end on mice that had been fasted 5–6 h and housed 31 days at the respective temperatures. Mice were challenged with 2 g/kg body weight. Area under the curve data (**L**) are presented as incremental data (iAUCs). Data are presented as means ± SEM. Dots indicate individual samples. *$P < 0.05$, **$P < 0.01$, **$P < 0.001$, ****$P < 0.0001$, $n = 7$.

these studies, the insulating effect of adiposity was minimal, but fur was found to provide 30–50% of the total insulation[4,24]. In dead mice, heat conductance, however, increased with ~450% immediately after death, suggesting that the insulating effect of fur requires operating physiological mechanisms, including vasoconstriction[24]. Besides species differences between mice and humans with regards to fur, the poor insulating effect of adiposity in mice is presumably also influenced by the consideration that the insulating factor of fat mass in humans is predominantly mediated by subcutaneous fat

mass (thickness)[26,27] which typically is less than 20% of total fat mass in rodents[28]. Furthermore, total fat mass may even be a suboptimal measure for insulation in humans since it has been argued that the increased surface area (and thereby increased heat loss) that inevitably follows with increased fat mass offsets the gain in insulation[6].

In the case of normal-weight mice, fasting plasma concentrations of TG, 3-HB, cholesterol, HDL, ALT and AST were notably unaffected by housing at different temperatures for almost five

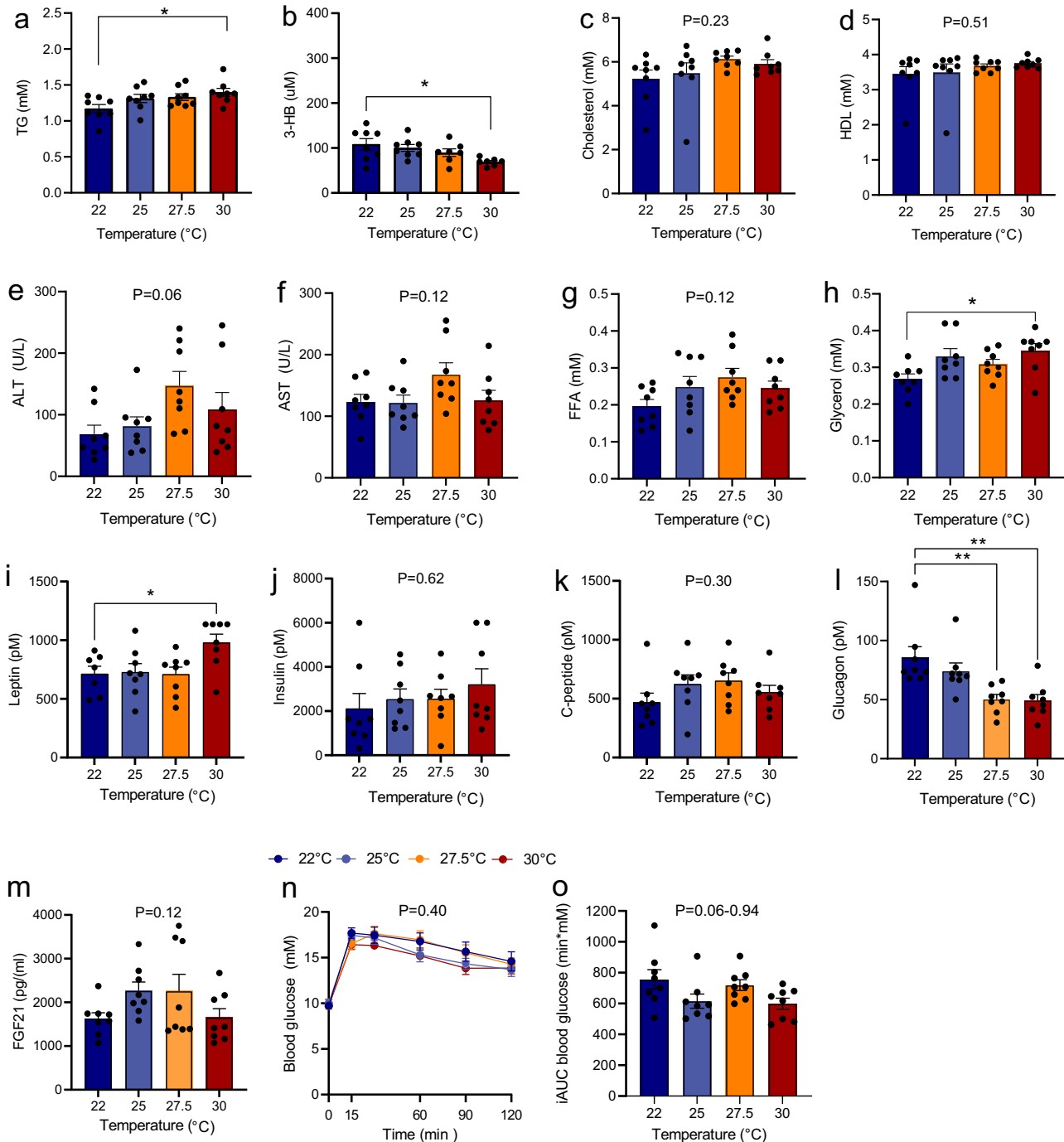

**Fig. 7 Effect of housing temperature on fasting plasma concentrations of lipids, ketone bodies, hepatic inflammatory markers, gluco- and appetite-regulating hormones, and glucose tolerance after an oral glucose.** Plasma concentrations of TG, 3-HB, cholesterol, HDL, ALT, AST, FFA, glycerol, leptin, insulin, C-peptide, glucagon, and FGF21 are shown in adult DIO male mice (**a–o**) after 33 days of housing at the indicated temperatures. Mice had been fasted 2–3 h prior to collection of blood. The oral glucose tolerance test is an exception to this as it was performed two days before study end on mice that had been fasted 5–6 h and housed 31 days at the respective temperatures Mice were challenged with 2 g/kg body weight. Area under the curve data (**o**) are shown as incremental data (iAUCs). Data are presented as means ± SEM. Dots indicate individual samples. *$P < 0.05$, **$P < 0.01$, **$P < 0.001$, ****$P < 0.0001$, $n = 7$.

weeks, presumably because mice were in similar states of energy balance and had similar weight and body composition at the end of the study. Consistent with the similarity in fat mass, plasma leptin was also not different, and fasting insulin, C-peptide and glucagon were also similar. In DIO mice, more signals were detected. Although mice at 22 °C in this case also had not overall been in a negative energy balance (since they gained weight) they

were at study termination in a relative more energy deficient situation compared to mice housed at 30 °C, as mirrored by an increased ketone body production (3-HB) and decreased plasma concentrations of glycerol and TG. The temperature-dependent differences in lipolysis did, however, not appear to be a consequence of intrinsic alterations (e.g., changes in adipose hormone-sensitive lipase expression) in epididymal or inguinal

fat, as FFA and glycerol release from extracted fat from these stores were similar between temperature groups. Although we did not investigate sympathetic tone in the current study, others have found that it is (based on heart rate and mean arterial pressure) linearly correlated with ambient temperature in the mouse and is about 20% lower at 30 °C than at 22 °C. Temperature-dependent differences in sympathetic tone may, therefore, have played a role for lipolysis in our study, but since increased sympathetic tone stimulates rather than inhibits lipolysis, other mechanisms may have counter-balanced such a potential effect to decrease lipolysis in mice housed at room temperature. Furthermore, part of the stimulatory effect of sympathetic tone on lipolysis is mediated indirectly by a strong inhibition on insulin secretion[29], underscoring the tonic break insulin has on lipolysis[30], but in our study fasting plasma insulin and C-peptide were similar, suggesting that potential differences in sympathetic tone across temperatures were not sufficient in effect size to alter lipolysis. Instead, we find it more likely that differences in energy status is the major driver of these differences in DIO mice. The underlying reason(s) that causes normal-weight mice to be better at adjusting food intake with EE warrants further investigation. However, overall, food intake is controlled by both homoeostatic and hedonic signalling[31–33]. Although it remains debated which of these two types of signalling are quantitatively more important[31–33], it is well recognized that prolonged intake of high-fat diet induces more pleasure-based eating behaviour and thereby to some extent uncouples homeostatic-regulated food intake[34–36]. Increased hedonic eating behaviour in DIO mice, that were fed 45% HFD, is therefore presumably part of the reason that these mice did not counter-balance food intake with EE. Interestingly, differences in appetite- and glucoregulatory hormones were also observed across housing temperature groups in DIO mice, but not in normal-weight mice. *In DIO mice*, plasma leptin was increased with temperature and glucagon was decreased with temperature. The extent to which temperature may directly have influenced these differences warrants further investigation, but in the case of leptin the relative negative energy balance and thus lower fat mass in the mouse at 22 °C has certainly played a significant role since fat mass and plasma leptin are highly correlated[37]. The explanation of the glucagon signal is, however, more puzzling. As in the case for insulin, glucagon secretion is powerfully inhibited by increased sympathetic tone[29], but the highest sympathetic tone would be expected to have been in the 22 °C group, which had highest plasma concentrations of glucagon. Another strong regulator of plasma glucagon is insulin, and insulin resistance and type-2-diabetes are strongly associated with both fasting and postprandial hyperglucagonemia[38,39]. However, the DIO mice in our study were equally insulin insensitive so this also cannot be the main driver of the increased glucagon signal in the 22 °C group. Hepatic fat content is also linked positively to increased plasma glucagon concentrations by mechanisms that may in turn involve hepatic glucagon resistance, decreased ureagenesis, increased circulating concentrations of amino acids and increased amino-acid-stimulated glucagon secretion[40–42]. However, as extractable concentrations of glycerol and TG did not differ between temperature groups in our study, this is presumably also not the underlying driver of the increased plasma concentrations in the 22 °C group. Triiodothyronine ($T_3$) plays a crucial role for overall metabolic rate and for initiating metabolic defence to prevent hypothermia[43,44]. Accordingly, plasma $T_3$ concentrations, presumably controlled by centrally mediated mechanisms[45,46], are increased in both mice and humans in conditions below thermoneutrality[47], although the increases in humans are smaller, which is consistent with mice having a greater propensity to lose heat to the environment. We did not measure plasma $T_3$ concentrations in the current study but

concentrations would likely have been lower in the 30 °C group, which could explain the effect on plasma glucagon in this group since we (Supplementary Fig. 5a) and others[48] have shown that $T_3$ increases plasma glucagon in a dose-dependent manner. Thyroid hormone has been reported to induce FGF21 expression in the liver[49]. Like glucagon, plasma concentrations of FGF21 also increase with increased plasma $T_3$ concentrations (Supplementary Fig. 5b and ref. [48]), but in contrast to glucagon, plasma FGF21 concentrations in our study were not affected by temperature. The underlying reason for this difference requires further investigation but it could be that a $T_3$-driven FGF21 induction needs to occur at a higher $T_3$ exposure level compared to the observed $T_3$-driven glucagon response (Supplementary Fig. 5b).

It has been shown that in mice housed at 22 °C, HFD is strongly associated with impaired glucose tolerance and (markers of) insulin-resistance[50]. However, when housed at thermoneutrality (defined here as 28 °C)[19], HFD is neither associated with impaired glucose tolerance nor with insulin-resistance. In our study, such a relationship was not reproduced in DIO mice, but normal-weight mice housing at 30 °C significantly improved glucose tolerance. The reason for this difference awaits further investigation but may be influenced by the consideration that the DIO mice in our study were insulin resistant with fasting plasma C-peptide and insulin concentrations 12–20 times higher than in the normal-weight mice and with fasting blood glucose concentrations of ~10 mM (was ~ 6 mM in the normal-weight group), presumably leaving a little window for any potential beneficial effects of housing at thermoneutrality to improve glucose tolerance. A possible confounder is that the OGTT, for practical reasons, was performed at room temperature. Mice that were housed at higher temperatures have, therefore, experienced a mild cold shock that may have impacted glucose absorption/clearance. However, based on the similar fasting blood glucose concentrations across temperature groups, the change in ambient temperature has presumably not influenced the outcome to a major extent.

As mentioned, it was recently highlighted that increasing housing temperature may alleviate some of the cold stress responses that could challenge translatability of mouse data to humans[9]. However, it remains unclear what the optimal housing temperature for mice is for mimicking human physiology. The answer to this question may also be influenced by field of research and the endpoint being investigated. Examples of this are impact of diet on liver fat accumulation, glucose tolerance, and insulin resistance[19]. In terms of energy expenditure, some investigators argue that since humans spend little extra energy to maintain core temperature, thermoneutrality is the optimal housing temperature, which they define for single housed adult mice as 30 °C[7,10]. Other researchers argue that an equivalent temperature to that normally experienced by humans for adult single housed adult mice is 23–25 °C, since they find thermoneutrality to be 26–28 °C and based on the argument that humans are about 3 °C below their lower critical temperature, defined here as about 23 °C when lightly clothed[8,12]. Our study concurs with several other studies which assert that thermoneutrality is not reached already at 26–28 °C[4,7,10,11,24,25], and thus provide evidence to show that housing at 23–25 °C is too low. With regards to housing temperature and thermoneutrality in mice, another important factor to consider is single housing vs. group housing. If mice are housed in groups rather than single housed as in our study, temperature sensitivity is reduced, presumably due to huddling. Nevertheless, when housed in groups of three, room temperature is still below the lower point of thermoneutrality[25]. Arguably, the most important inter-species difference in this regard is the quantitative importance of BAT activity as a defence against hypothermia. Thus, whereas mice largely compensate for their higher heat-loss by upregulating the activity of BAT, accounting alone for over 60%

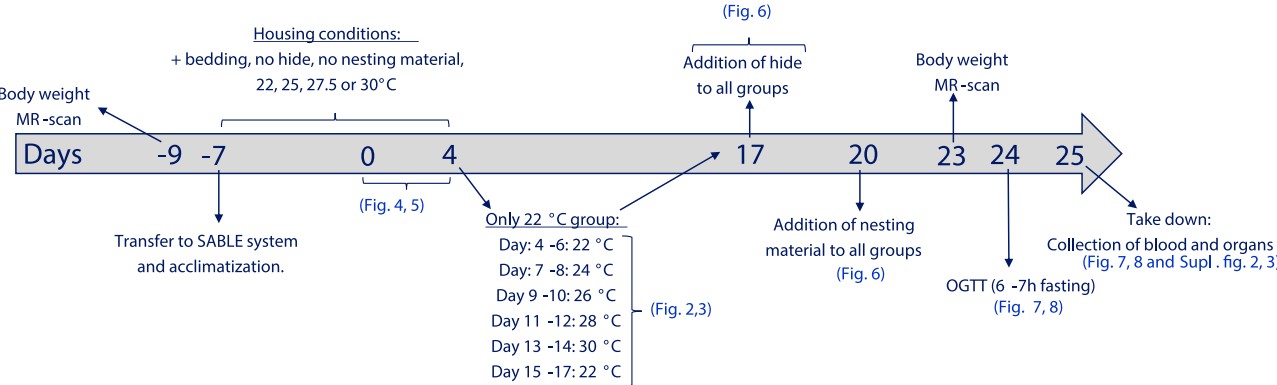

**Fig. 8 Study design.** Normal-weight and DIO mice followed same study procedure. At day -9, mice were weighted and MR-scanned and allocated into groups matched on weight and body composition. At day -7 mice were transferred into a closed temperature controlled indirect calometry system from SABLE Systems International (NV, USA). Mice were single housed with bedding but without hide and nesting material. The temperature was either set to 22, 25, 27.5, or 30 °C. After a week of acclimatization (day -7 to day 0, no handling of animals), data was collected for four consecutive days (day 0–4, data presented in Figs. 1, 2, 5). Hereafter, the mice housed at 25, 27.5, and 30 °C was left at unaltered conditions until day 17. Meanwhile, the temperature in the 22 °C group was increased in 2 °C intervals every other day, adjusting temperature at start of light cycle (0600 h) (data are presented in Fig. 1). At day 15, temperature was lowered to 22 °C and data was collected for two days to provide a baseline data for following procedures. At day 17, a hide was added to all mice, and nesting material was added at day 20 (Fig. 5). At day 23, mice were weighted and MR-scanned and were thereafter left undisturbed for 24 h. At day 24, mice were fasted from start of light cycle (0600 h) and were subjected to an OGTT (2 g/kg) at 1200 h (6–7 h fasted). Mice were thereafter returned to their respective housing conditions in the SABLE system and were euthanized the following day (day 25).

of the EE when housed at 5 °C[51,52], the contribution of human BAT activity to EE is considerably smaller[53]. Reducing BAT activity may therefore be an important way to increase human translation. Regulation of BAT activity is complex but are generally regulated by combined actions of adrenergic stimulation, thyroid hormones, and UCP1 expression[14,54–57]. Our data shows that temperature needs to be raised to above 27.5 °C to detect differences in BAT expression of genes responsible for function/activation compared to mice at 22 °C. The differences detected between the 30 and 22 °C groups did, however, not uniformly point towards upregulation of BAT activity in the 22 °C group, since *Ucp1*, *Adrb2*, and *Vegf-a* were downregulated in the 22 °C group. The underlying reason for these unexpected findings remains to be identified. One possibility is that their increased expression may not reflect a signal of housing at warmer temperature but could rather be an acute effect of moving them from 30 to 22 °C at the day of take-down (mice experienced room temperature for 5–10 min before take-down).

A general limitation of our study is that we only investigated male mice. Other studies have demonstrated that sex may be an important factor for consideration in our primary readouts since single housed female mice is more temperature sensitive due to higher heat conductance and more tightly controlled maintenance of core temperature[25]. Furthermore, female mice (on HFD) had greater coupling of energy intake with EE at 30 °C compared to male mice which overconsumed more and thus, in agreement with our results, gained significant more weight compared to sex matched mice housed below thermoneutrality (in this case 20 °C)[20]. Effects of housing below thermoneutrality was, therefore, higher in female mice[25], but followed the same pattern as in male mice[25]. In our study, we focused on single-housed male mice since this is the condition most metabolic studies investigating EE are performed under. Another limitation of our study is that mice were on the same respective diets for the entirety of the studies, precluding investigation of the importance of housing temperature for metabolic flexibility (as measured by change in RER in response to change in diets with different macronutrient composition). Metabolic flexibility has, however, been reported to be higher in both female and male mice housed at 20 °C compared to matched mice housed at 30 °C[20].

In summary, our studies shows, consistent with other studies[4,7,10,11], that in single housed normal-weight mice thermoneutrality is above the suggested 27.5 °C. Furthermore, our study indicates that adiposity does not confer a major insulating factor in either normal-weight or DIO mice, resulting in similar temperature: EE relationship in DIO and normal-weight mice. Whilst normal weight mice matched food intake with EE, and thus remained weight stable across housing temperatures, food intake in DIO mice were the same across temperatures, causing mice at 30 °C to gain more weight than mice at 22 °C. In general, systematic studies investigating the potential importance of housing at temperatures below thermoneutrality for the often-observed poor translatability between mouse studies to humans is much warranted. For instance, within obesity research, part of the explanation of the general poor translatability may be related to the circumstance that weight loss studies in mice are usually performed in mildly cold-stressed animals housed at room temperature, which, due to their increased EE, may show exaggerated weight loss compared to that anticipated in humans; in particular, if the mode of action depends on elevated EE by increasing BAT activity, which is more active and more activatable at room temperature than at 30 °C.

## Methods

**Ethical considerations**. Animal studies were conducted with permission from the Danish Animal Experiments Inspectorate (2020-15-0201-00683) in accordance with the guidelines of Danish legislation governing animal experimentation (1987) and the National Institutes of Health (publication number 85-23) and the European Convention for the Protection of Vertebrate Animals used for Experimental and other Scientific Purposes (Council of Europe No 123, Strasbourg 1985).

**Animals, housing, and pre-experimental procedures**. Male C57BL/6J mice at 20 weeks of age were obtained from Janvier Saint Berthevin Cedex, France and housed with *ad libitum* access to standard chow (Altromin 1324) and water, following a 12:12 h light:dark cycle at room temperature (~22 °C). DIO male mice (20 weeks) were obtained from the same supplier and were housed with *ad libitum* access to 45% high-fat diet (Cat no. D12451, Research Diet Inc. NJ, USA) and water. Mice were acclimated to their environment for one week before study initiation. Two days prior to transfer into the indirect calorimetry system, mice were weighed, MR-scanned (EchoMRI™, TX, USA), divided into four groups, matched on body weight, fat, and normal-weight mass.

**Indirect calorimetry in adult normal-weight male mice**. A graphical outline of the study design is shown in Fig. 8. Mice were transferred into a closed and temperature adjustable indirect calorimetry system from Sable Systems Internationals (NV, USA) which included mass monitors for food and water and Promethion BZ1 frames recording activity level by measuring XYZ beam breaks. Mice ($n = 8$) were single-housed at either 22, 25, 27.5, or 30 °C with bedding material but without a hide and nesting material, following a 12:12 h light:dark cycle (light: 0600-1800h). Air flow was 2500 ml/min. Mice were acclimatized for seven days before recordings were started. Recordings were collected for four consecutive days. Thereafter, mice at 25, 27.5, and 30 °C remained at their respective temperatures for another 12 days before adding cage enrichments as described below. In the meantime, the group of mice housed at 22 °C were left at this temperature for two more days (for collection of new baseline data), thereafter temperature was increased in 2 °C increments every other day at start of light phase (0600 h) until reaching 30 °C. Afterwards, temperature was decreased to 22 °C and data was collected for another two days. Following another two days of recording at 22 °C, hides were added to all cages at all temperatures, and data collection was started the following day (day 17) and throughout three days. Hereafter (at day 20), nesting material (8–10 g) was added to all cages at start of light cycle (0600), and data was collected for another three days. At end of the study, mice housed at 22 °C had thus been at this temperature for 21/33 days and at 22 °C the last 8 days whereas mice at the other temperatures had been at this temperature for 33/33 days. Mice remained of chow during the study.

**Indirect calorimetry in DIO male mice**. DIO mice ($n = 8$) followed the same protocol as the normal-weight mice (described above and in Fig. 8). Mice remained on 45% HFD for the entirety of the energy expenditure experiments.

**Recordings and data processin**. VO$_2$ and VCO$_2$ and water vapour pressure were recorded at 1 Hz with a 2.5 min cage time constant. Food intake and water intake was collected by continuous recording (1 Hz) of weights of food and water hoppers. Mass monitors used have a reported resolution of 0.002 g. Activity levels were recorded using three-dimensional XYZ beam array monitors, collecting data with an internal resolution of 240 Hz and reporting it every second, to quantify total distance travelled (m) with a reported effective spatial resolution of 0.25 cm. Data was processed by Sable Systems Macro Interpreter v.2.41, calculating EE and RER and filtering out abnormal values (e.g., erroneous food intake events). The Macro Interpreter was set to output data at five-minute intervals for all parameters.

**Importance of ambient temperature for glucose tolerance in normal-weight and DIO male mice**. Besides regulating EE, ambient temperature may also regulate other aspects of metabolism, including post-prandial glucose metabolism, possibly through the regulation of the secretion of glucometabolic hormones. To test this hypothesis, we finalized the temperature study in both normal-weight and DIO mice by challenging them with an oral glucose load (2 g/kg). Methods are described in detail in Supplementary Material.

**Importance of ambient temperature for lipids and hormones in plasma and lipids in the liver**. At end of study (day 25), mice were fasted from 2 to 3 h (from 0600 h), were anesthetised with isoflurane and were total bleed by retroorbital vein puncture. Quantification of plasma lipids and hormones and lipids in the liver were performed are described in Supplementary Material.

**Ex-vivo lipolysis assay**. To investigate whether housing temperature leads to intrinsic changes in fat tissue that affect lipolysis, inguinal and epididymal fat tissue were excised from mice directly after the terminal bleeding. Tissue was processed in a newly developed ex-vivo lipolysis assay described in Supplementary Methods.

**Expression of thermogenic genes in brown adipose tissue**. Brown adipose tissue (BAT) was harvested on the day of study termination and was processed as described in supplementary methods.

**Statistics and reproducibility**. Data are presented as means ± SEM. Graphs were made in GraphPad Prism 9 (La Jolla, CA) and figures were edited in Adobe Illustrator (Adobe Systems Incorporated, San Jose, CA). Statistical significance was assessed in GraphPad Prism and tested by either paired t-test, One-/two-way ANOVA for repeated measurements followed by Tukey's multiple comparison test, or unpaired one-way ANOVA followed by Tukey multiple comparison test, as appropriate. Gaussian distribution of data was confirmed prior to testing by the D'Agostino-Pearson normality test. Sample sizes are indicated in respective parts of the result section as well as in figure legends. Replicates were defined as any measurements performed on the same animal (in vivo or on tissue samples). Regarding data reproducibility, the coupling between energy expenditure and housing temperature has been confirmed in four separate studies with similar study design using different mice.

**Resource availability**. Detailed experimental protocols, materials, and raw data are available upon reasonable request to lead author Rune E. Kuhre. This study did not generate new unique reagents, GMO animals/cell lines, or sequencing data.

**Reporting summary**. Further information on research design is available in the Nature Research Reporting Summary linked to this article.

## Data availability

All data forming Figs. 1–7 has been deposited at Science Data Bank repository, accession number: 1253.11.sciencedb.02284 or https://doi.org/10.57760/sciencedb.02284. Data presented in ESM can be send upon reasonable test to Rune E Kuhre.

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

## Acknowledgements

We would like to express our gratitude to animal technicians Helle Andersen and Maja Gross Christensen for technical assistance (Global Obesity and Liver Disease, Global Research, Novo Nordisk A/S, Måløv, Denmark), as well as the animal technicians in Novo Nordisk Animal unit (Global Discovery and Development, Novo Nordisk A/S, Måløv, Denmark) for general care-taking. Furthermore, we would like to express our gratitude to Susanne Jørgensen and Johannes Josef Fels (Research Bioanalysis, Global Research Technologies, Novo Nordisk A/S, Måløv, Denmark) for quantification of plasma hormone concentrations.

## Author contributions

R.E.K. concepted the study. L.M.R., N.P., M.K.G., L.T., K.P., B.Q.C., and R.E.K. designed and performed studies. L.M.R., N.P., M.K.G., L.T., K.P., B.Q.C., and R.E.K. analyzed and interpreted data. L.M.R. and R.E.K. drafted the manuscript. N.P., M.K.G., L.T., K.P., and B.Q.C. revised the manuscript and provided important intellectual content.

## Competing interests

The authors declare the following competing interest: all authors are employed by Novo Nordisk (Denmark), and some are minor shareholders of Novo Nordisk stocks. All authors declare no competing interests that may be of relevance to this work.
