## [Peer Review File · Communications Biology]

Reviewers' comments:

Reviewer #1 (Remarks to the Author):

The paper from John et al. entitled "Housing temperature reveals that energy intake counter-balances energy expenditure in normal-weight mice, but are only weakly coupled in DIO mice" attempts to examine the impact of housing temperature on the interaction of energy intake and energy expenditure in normal and diet-induced obese mice. Specifically, the authors set out to '...characterize the impact of ambient temperature on EE, energy homeostasis, and glycemc control in normal weight and DIO mice.'

Overall, the data presented in this manuscript is quite interesting and relevant to the ongoing debate regarding mouse housing temperature in pre-clinical biomedical research. However, the authors present the data in a manner that complicates reader comprehension, have not completely analyzed the wealth of data collected, or considered how this data relates to recent findings in similar studies.

Comments are listed below:

Major

Methods

- C57Bl/6 sub-strain information is necessary. This is important for the reader to rationalize observed differences in 22C and 30C that are not as large as previously observed (e.g. - ~40%).
- The authors appear to have designed an excellent acclimation plan for the indirect studies. However, the methods text is not clear. Early in section 2.2 the authors state that the animals '...were acclimated to their environment for two weeks.' From the supplemental schematic, it appears that they were actually acclimated to temperature and indirect caging for seven days. Further, the description of the differences in data collection between in the 22C mice and the other temperatures is unclear.
- Sample size should also be stated in the methods and listed in each figure legend.
- In all Sable systems data from the MM1s and beams is collected every second. Additionally, a description of how the food, water, and activity data is collected and analyzed is necessary.
- The authors need to describe that the DIO mice remained on HFD for the entirety of the indirect experiments.
- OGTT – The temperature of the room where the GTTs were performed is not described. While limited data are available describing the impact of ambient temperature on mouse glucose homeostasis, if the animals did not undergo the GTT at the appropriate experimental temperature then this data is open to question. No description of dosing normalization (kg of what...body weight or FFM), or time points collected. Also, the text says the fast was from 5-6 hrs the supplement says 6-7 hrs.
- Describe where the plasma came from for the analysis listed in section 2.7. The authors just finished describing a terminal blood sample at 120 minutes into the OGTT. It is unclear which plasma is used in section 2.7.
- Were the ex vivo lipolysis samples agitated or rocked during the described incubations? Was a CO2 cell culture incubator used?
- Concerns....1) small n-size, 2) time spent at RT prior to euthanasia and sample collection, 3) no BAT westerns.

Results

- The longitudinal representation of the 1hr binned data in Figure 1 - 3 (supplement figure 4) is visually distracting and adds no insight to any of the outcomes for which it is displayed, as the authors make no mention of potential differences in the light or dark cycle in these result sections. Again, 12hr data would still be more clearly rendered in bar graph form. (The only possible exception would be the RQ data and the authors make no mention of metabolic flexibility in the manuscript). Removing the 1 hr binning would create a much cleaner and more straightforward presentation of the data to the reader.
- Figure 1. In the methods supplement the authors describe 48 hrs spent at 24, 26, 28, & 30C, but graphically represent this as 24 hrs. It is assumed that the data represented is the last 24hrs of the two day data collection. Describe that in the figure legend and methods section.
- Panels A – C in Figures 2 & 3, are repetitive. The authors display the same data in subsequent panels later in the figures.

- The authors state that significant differences were observed for reducing EE following addition of a hide/bedding at 27.5C. No significance is annotated in Figure 4F.
- The placement of the data relating to the assessment of insulation after the blood analysis is confusing.
- Figure 6. While the calculation of insulation as ratio of ambient temperature to EE has precedent, the Pearson coefficients relative to body composition are questionable. The body composition data was collected either 9 days before the EE data or 6 days afterward. Along with the low sample size, this discrepancy in data collection could explain a large portion of the variability observed.
- The authors have the data available to calculate energy balance yet do not present this potentially valuable data. This is important considering the authors assertions that energy intake is 'counter-balanced' by EE in normal mice but not DIO mice, and that housing temperature impacts energy homeostasis. Importantly, Morris et al. (Obesity 2020) recently observed a similar phenotype in acute HFD feeding.
- The authors should consider the obvious metabolic inflexibility in the respiratory quotient data.

Major Weaknesses: 1) No females. The authors have no justification for not performing these experiments in female mice. However, the authors can simply state this as a weakness in the text. 2) The authors fail to cite pertinent recent literature: Morris et al. Obesity 2020, Ono-Moore et al. AJP Endo 2020, Skop et al. Cell Rep 2020, Abreu-Vieira et al. Mol Metab 2015.

Reviewer #2 (Remarks to the Author):

In this paper, the authors further the discussion concerning the "correct" environmental temperature for making mouse experiments relevant for human outcome. This is, particularly for the translational aspects of therapeutic advances, an important issue, and I believe that detailed and competent analysis of this aspect should be promoted. Particularly, there has been a discussion in the literature related to the question: if it is accepted that 20 °C is too low a temperature, which temperature above 20 °C is then "better", and this is the question the authors here address. I am thus in general positive to making these data available by publishing them, but I think presentation-wise the paper could be substantially improved.

I have a series of points of different dignity:

1) section 2.2: the environmental temperature here, before the real experiment, is not stated.

2) The lay-out of the experiments is difficult to follow and I would prefer that suppl fig 1 was included in the main figs.

3) I am not directly aware of the limitations for this journal but the authors have combined so much in each fig that it becomes very difficult to see them, even worse so in a journal edition. I would at least divide the data from normal and DIO mice into different figs, provided the journal does not have limitations.

4) In general, I do not approve of exaggerating y-axes. Thus, there are clear effects of temperature but they are not really as large as presented in fig 1B etc. I am in strong favour of showing this type of data with a 0-starting y-axis. This will give a correct picture of the relative effect and also demonstrate that the single values are very close, much closer than seen on the present figs. This of course goes also for fig 1L – but here one could add that the authors by changing the top of the y-axes also diminish the impression of difference between the chow and the DIO in absolute values (which is quite remarkable). So perhaps y-axes from 0 to 16 for both would be a better representation of these results. Similar comments concerning 2O, 3A (perhaps not a necessary fig), 3C, 3N, 3P. The exception is evidently the RER graphs where a relevant y-axis should be kept – but it should be consistent in all figs, e.g. 0.5 to 1.2.

5) In the Results section, the authors write as if the figs don't exist. Thus, we get all the values shown in the figs, and all the stats shown in the figs. There is no point in this, and all these values should be removed from the Results section; they make it unreadable. Statements such as "30 %

lower at 30 °C" should of course be kept. Although there are ideas out there that Results should be shown "by themselves" and not directly commented, I think this is meaningless, given how experiments are planned and conducted. The authors should use the gained space to make primary comments on their results (some of this can be moved in from present discussion); e.g. comment at the end of 3.1.2 on the differences between chow and DIO.

6) Fig 2E should look like 2I and 2M, but just leave the 30 °C value bar empty. The comment on the technical problem should also be in the fig legend.

7) line 345: It is a little difficult what the authors want to say with the first statement here – but the "however" in the next sentence seem to imply that in humans (that admittedly do not have fur), adiposity increases insulation. However, when I read the abstract from Nahon it says: "We conclude that larger body sizes possessed reduced LCT as explained by higher BMR related to more lean mass rather than a change in whole-body conductance. Thus, larger individuals with higher lean mass need to be exposed to colder temperatures to activate CIT, not because of increased insulation, but because of a higher basal heat generation. " The present authors should thus avoid given the impression that mice and humans are different in this respect, according to the data of Nahon.

Line 350: what does "same" mean here? And on the fig? I may start to understand – I believe these data are from fig 1 – but what is the °C? For normal calculation of heat loss, I would agree on EE/°C – i.e. the slope of a plot for one mouse where EE was expressed as a function of ambient temperature. I guess then that insulation would be the opposite, °C/EE – but I cannot see the sense in the present representation. If I understand it correctly, the authors should calculate the slope for each mouse and plot that versus fat mass. They could probably plot the data from chow and DIO mice on the same graph. – Concerning the "different" mice, it gets further complicated. As far as I can see, the only choice is to plot the mean EE for all mice as a function of ambient temperature and get a single value for a single mean value of fat mass – and do the same for the DIOs. Thus, all of EG comes out as only two points (!) (similar for FH), and even that is somewhat problematic. OK – this is how I understand it; if the authors have different definitions of insulation etc. the present representations may be adequate but as it is, this fig is not meaningful to me.

Fig 4: it is remarkable here that the difference between EE at 30 °C and 22 °C is now more than 100 % (left bar in H versus left bar in B). This should at least be commented.

Fig 5KL: I do not in general consider it correct to calculate AUC in GTT by the "total" AUC – the initial (fasting) glucose level then dominates the outcome. Here there is no difference in initial glucose level – but the effect of temperature is diminished by this way of calculating. The authors should at least also show a plot of the incremental AUC (the change from initial level).

However, the above comments (except for the incomprehensible insulation data) do not take away from the main message of the paper, and with adequate modifications I think this paper will become acceptable for publication.

Response letter to COMMSBIO-22-0446-T

Reviewer #1 (Remarks to the Author):

The paper from John et al. entitled “Housing temperature reveals that energy intake counter-balances energy expenditure in normal-weight mice, but are only weakly coupled in DIO mice” attempts to examine the impact of housing temperature on the interaction of energy intake and energy expenditure in normal and diet-induced obese mice. Specifically, the authors set out to ‘...characterize the impact of ambient temperature on EE, energy homeostasis, and glycemic control in normal weight and DIO mice.’

Overall, the data presented in this manuscript is quite interesting and relevant to the ongoing debate regarding mouse housing temperature in pre-clinical biomedical research. However, the authors present the data in a manner that complicates reader comprehension, have not completely analyzed the wealth of data collected, or considered how this data relates to recent findings in similar studies.

Dear reviewer. We sincerely appreciate your overall positive assessment of our work as well as suggestions for improvements. As you will appreciate from the point-to-point responses below, we have acted on all or your points.

Comments are listed below:

Major

Methods

- C57Bl/6 sub-strain information is necessary. This is important for the reader to rationalize observed differences in 22C and 30C that are not as large as previously observed (e.g. - ~40%).

We agree and have added this information to the material and methods section. “Male C57BL/6J mice at 20 weeks”.

- The authors appear to have designed an excellent acclimation plan for the indirect studies. However, the methods text is not clear. Early in section 2.2 the authors state that the animals ‘...were acclimated to their environment for two weeks.’ From the supplemental schematic, it appears that they were actually acclimated to temperature and indirect caging for seven days. Further, the description of the differences in data collection between in the 22C mice and the other temperatures is unclear.

Thanks for pointing this ambiguity to our attention. Mice were, as indicated in the study design figure (supl. Fig. 1 in first version submitted, now fig. 1), acclimated for seven days before recordings were started. This is now updated in section 2.2.

- Sample size should also be stated in the methods and listed in each figure legend.

We agree and have now included this information.

- In all Sable systems data from the MM1s and beams is collected every second. Additionally, a description of how the food, water, and activity data is collected and analyzed is necessary.

We agree and have updated this section to following:

Recordings and data processing: VO_2 and VCO_2 and WVP were recorded at 1Hz with a 2.5-minute cage time constant. Food intake and water intake was collected by continuous recording (1 Hz) of weights of food and water hoppers. Mass monitors used have a reported resolution of 0.002 g. Activity levels were recorded using three-dimensional XYZ beam array monitors, collecting data with an internal resolution of 240Hz and reporting it every second, to quantify total distance travelled (m) with a reported effective spatial resolution of 0.25 cm. Data was processed by Sable Systems Macro Interpreter v.2.41, calculating EE and RER and filtering out abnormal values (e.g. erroneous food intake events). The Macro Interpreter was set to output data at five-minute intervals for all parameters.

- The authors need to describe that the DIO mice remained on HFD for the entirety of the indirect experiments.

We agree and have added this information to section 2.4. Indirect calorimetry in DIO male mice: "... Mice remained on 45% HFD for during the entirety of the energy expenditure experiments".

- OGTT – The temperature of the room where the GTTs were performed is not described. While limited data are available describing the impact of ambient temperature on mouse glucose homeostasis, if the animals did not undergo the GTT at the appropriate experimental temperature then this data is open to question. No description of dosing normalization (kg of what...body weight or FFM), or time points collected. Also, the text says the fast was from 5-6 hrs the supplement says 6-7 hrs.

This is a good point. OGTT's were performed at room temperature and dosing's were based on body weight. These information's have now been added to the M&M, section 2.6 (additions are underlined). Fasting time was 6-7h. The text has been updated accordingly.

"To test this hypothesis, we finalized the temperature study in both normal-weight and DIO mice by challenging them with an oral glucose load (2 g/kg body weight, 50% (w/v) glucose, 4 ml/kg). At this stage, mice had been housed at least 8 days at the respective temperatures. Mice were fasted for 6-7h (food was removed at start of light cycle (0600h)) and were placed in their cages in a procedure room (22°C) two hours before the challenge to reduce the influence of acute responses to changes in ambient temperature".

We acknowledge that this acute change in ambient temperature for some of the mice may have confounded their glucose tolerance. We now mentioned this limitation in the discussion section:

"A possible confounder is that the OGTT, for practical reasons, was performed at room temperature. Mice that were housed at higher temperatures have, therefore, experienced a mild cold shock that may have impacted glucose absorption/clearance. However, based on the similar fasting blood glucose concentrations across temperature groups, the change in ambient temperature have presumably not influenced the outcome to a major extent".

- Describe where the plasma came from for the analysis listed in section 2.7. The authors just finished describing a terminal blood sample at 120 minutes into the OGTT. It is unclear which plasma is used in section 2.7.

Thank you for pointing out that details of this was sparse. We have now updated the section “2.6” and section “2.7”. In the process, we realized that we have made an unfortunate mistake; terminal blood samples for plasma hormone and lipid analysis were not withdrawn until the day after the OGTT (as indicated in the study design in Suppl. Figure 1.

- Were the ex vivo lipolysis samples agitated or rocked during the described incubations? Was a CO2 cell culture incubator used?

We agree that this information should have been added in the initial version of the manuscript. We have added this information in the relevant M&M section in the revised, here shown with underlined writing:

“Hereafter, the buffer was replaced with fresh stimulation buffer and incubated for 1 hour. Glycerol and FFA content were quantified as described in section 2.7 and were normalized to tissue weight (g). Plates were shaken on an orbital shaker (low speed 150 rpm) for 20 s after medium addition and before the collection”.

- Concerns....1) small n-size, 2) time spent at RT prior to euthanasia and sample collection, 3) no BAT westerns.

These are all valid concerns. We agree that an n=7-8 is a low sample size for some types of in vivo studies. However, inter-animal variability on the primary endpoints (energy expenditure and food intake) was low, as also evident from the data. We therefore find it unlikely that a larger sample size would have yielded different results. Time spend at RT before euthanasia was reduced as much as possible and was ≤ 10 min.

Results

- The longitudinal representation of the 1hr binned data in Figure 1 - 3 (supplement figure 4) is visually distracting and adds no insight to any of the outcomes for which it is displayed, as the authors make no mention of potential differences in the light or dark cycle in these result sections. Again, 12hr data would still be more clearly rendered in bar graph form. (The only possible exception would be the RQ data and the authors make no mention of metabolic flexibility in the manuscript). Removing the 1 hr binning would create a much cleaner and more straightforward presentation of the data to the reader.

We agree that the 1hr binned data are difficult to assess quantitatively. However, at the same time we think they should be included in the data presentation as they are indicative of general data quality. The presentation of data in the results section and discussion are exclusively based on the 24h bar graphs and as such, we do not think removal of the 1hr data would increase readability to a significant extent.

- Figure 1. In the methods supplement the authors describe 48 hrs spent at 24, 26, 28, & 30C, but graphically represent this as 24 hrs. It is assumed that the data represented is the last 24hrs of the two day data collection. Describe that in the figure legend and methods section.

Answer: Yes this is correct, and we agree that this should be detailed. In supplementary figure 1 (now figure 1 in main document), it is indicated that mice that initially were housed at 22°C and experienced gradual increasing housing temperatures in 2°C increments.

- Panels A – C in Figures 2 & 3, are repetitive. The authors display the same data in subsequent panels later in the figures.

We agree that A-C are repetitive in figure 2 (now figure 4) and have deleted these figures from the revised figure. In figure 3 (now figure 5) they are, however, not repetitive and have thus stayed in the figure.

- The authors state that significant differences were observed for reducing EE following addition of a hide/bedding at 27.5C. No significance is annotated in Figure 4F.

This is a mistake, which we have now corrected. No differences were detected in the 27.5°C group. Thank you for bringing this to our attention.

- The placement of the data relating to the assessment of insulation after the blood analysis is confusing.

We agree. Considering the technical shortcomings of the insulation analysis, as the reviewer points out in the next point, we have decided to omit the data from our paper.

- Figure 6. While the calculation of insulation as ratio of ambient temperature to EE has precedent, the Pearson coefficients relative to body composition are questionable. The body composition data was collected either 9 days before the EE data or 6 days afterward. Along with the low sample size, this discrepancy in data collection could explain a large portion of the variability observed.

We agree that this is a limitation of our study that body composition data was collected at a different time than EE data. Considering this the uncertainty with regards to how data may have been influenced by this, we have decided to remove these data.

- The authors have the data available to calculate energy balance yet do not present this potentially valuable data. This is important considering the authors assertions that energy intake is ‘counter-balanced’ by EE in normal mice but not DIO mice, and that housing temperature impacts energy homeostasis. Importantly, Morris et al. (Obesity 2020) recently observed a similar phenotype in acute HFD feeding.

Many thanks for pointing the Morris study to our attention. We agree that the findings in this study aligns well with our findings and have mentioned this in the discussion section with following:

“Furthermore, female mice (on HFD) had greater coupling of energy intake with EE at 30°C compared to male mice which overconsumed more and thus, in agreement with our results, gained significantly more weight compared to sex matched mice housed below thermoneutrality (in this case 20°C)²³”.

With regards to the suggestion of calculating energy balance we find it relevant and are doing this in a follow up study that are designed at investigate the importance of housing temperature for metabolic flexibility. In present work, it will be difficult to fit more data. We therefore prefer the current assessment of long-standing energy homeostasis based on dynamics in body weight.

- The authors should consider the obvious metabolic inflexibility in the respiratory quotient data.

We assume that this comment concerns the DIO mice, since the normal weight mice on chow do show diurnal dynamics in RER. However, since normal-weight mice and DIO mice were on different diet, comparison of these two groups with regards to metabolic flexibility is not straightforward given that diet would be a confounding factor. We agree that it is important to investigate this further, but such an analysis requires another study design where different diets are supplied to the same mice, e.g. as in the study by Morris EM, et al. (Obesity, vol. 12, 2020). We thus added following to the discussion:

..." Another limitation of our study is that mice were on the same respective diets for the entirety of the studies, precluding investigation of the importance of housing temperature for metabolic flexibility (as measured by change in RER in response to change in diets with different macronutrient composition). Another study found that metabolic flexibility was higher in both female and male mice housed at 20°C compared to matched mice housed at 30°C²³."

Major Weaknesses:

1) No females. The authors have no justification for not performing these experiments in female mice. However, the authors can simply state this as a weakness in the text.

We agree that it is a general limitation that our study is only performed in male mice, since sex may have an impact on recently highlighted in paper by Skop V et al. (Pubmed ID: 34478905). We have mentioned this limitation in the discussion and elaborated more on the subject as shown in the underlined text below.

"A general limitation of our study is that we only investigated male mice. Other studies have demonstrated that sex may be an important factor for consideration in our primary readouts since single housed female mice is more temperature due to higher heat conductance and more tightly controlled maintenance of core temperature²⁸. Furthermore, female mice (on HFD) had greater coupling of energy intake with EE at 30°C compared to male mice which overconsumed more and thus, in agreement with our results, gained significant more weight compared to sex matched mice housed below thermoneutrality (in this case 20°C)²³. Effects of housing below thermoneutrality was, therefore, higher in female mice²⁸, but followed the same pattern as in male mice²⁵."

Furthermore, we have added the following limitations.

"Another limitation of our study is that mice were on the same respective diets for the entirety of the studies, precluding investigation of the importance of housing temperature for metabolic flexibility (as measured by change in RER in response to change in diets with different macronutrient composition). Another study found that metabolic flexibility was higher in both female and male mice housed at 20°C compared to matched mice housed at 30°C²³."

2) The authors fail to cite pertinent recent literature: Morris et al. Obesity 2020, Ono-Moore et al. AJP Endo 2020, Skop et al. Cell Rep 2020, Abreu-Vieira et al. Mol Metab 2015.

Many thanks for pointing these relevant studies to our attention. We have now included all of them in the relevant places.

Reviewer #2 (Remarks to the Author):

In this paper, the authors further the discussion concerning the “correct” environmental temperature for making mouse experiments relevant for human outcome. This is, particularly for the translational aspects of therapeutic advances, an important issue, and I believe that detailed and competent analysis of this aspect should be promoted. Particularly, there has been a discussion in the literature related to the question: if it is accepted that 20 °C is too low a temperature, which temperature above 20 °C is then “better”, and this is the question the authors here address. I am thus in general positive to making these data available by publishing them, but I think presentation-wise the paper could be substantially improved.

Dear reviewer. Many thanks for the overall positive and thorough assessment of our work. Please find below our point-point response to your insightful and constructive points raised, which collectively improves our work substantially.

I have a series of points of different dignity:

1) section 2.2: the environmental temperature here, before the real experiment, is not stated.

Good point. The mice were prior to acclimatization in the SABLE system housed at std. conditions (22°C). This information is now added in the section 2.2.

“Male C57BL/6J mice at 20 weeks of age were obtained from Janvier Saint Berthevin Cedex, France and housed with *ad libitum* access to standard chow (Altromin 1324) and water, following a 12:12 h light:dark cycle at room temperature ($\approx 22^{\circ}\text{C}$)”.

2) The lay-out of the experiments is difficult to follow and I would prefer that suppl fig 1 was included in the main figs.

We agree that the experimental design is rather complex. The point about including current supplementary figure as fig. 1 in the main text is well taken. We have therefore changed this accordingly.

3) I am not directly aware of the limitations for this journal but the authors have combined so much in each fig that it becomes very difficult to see them, even worse so in a journal edition. I would at least divide the data from normal and DIO mice into different figs, provided the journal does not have limitations.

This is a very good point and a point we also were worried about. The journal allows up to ten display items for a paper of our length. We have, therefore, followed your recommendation and divided the combined data from normal weight mice and DIO mice in fig. 1 and 5 in into separate figures. With the addition of study design now being fig. 1 in the main document, that takes us to 9 figures in total in main document (and four figures in supplementary document).

4) In general, I do not approve of exaggerating y-axes. Thus, there are clear effects of temperature, but they are not really as large as presented in fig 1B etc. I am in strong favour of showing this type of data with a 0-starting y-axis. This will give a correct picture of the relative effect and also demonstrate that the single values are very close, much closer than seen on the present figs. This of course goes also for fig 1L – but here one could add that the authors by changing the top of the y-axes also diminish the impression of difference between the chow and the DIO in absolute values (which is quite remarkable). So perhaps y-axes from 0 to 16 for both would be a better representation of these results. Similar comments concerning 2O,

3A (perhaps not a necessary fig), 3C, 3N, 3P. The exception is evidently the RER graphs where a relevant y-axis should be kept – but it should be consistent in all figs, e.g. 0.5 to 1.2.

This is a reasonable and worthwhile objection. We have thus changed the y-axes they now all starts at 0, except for RER data. With regards to RER-values, we respectfully disagree to the suggestion of standardization of Y-axis between normal weight and DIO animals since mice are on different diets (chow vs. 45% HFD) and therefore have different RER-values and size of daily fluctuations. However, we agree that it is sensible to present different RER-values from respectively normal-weight and DIO mice in a uniform way and thus standardized the axes across figures.

5) In the Results section, the authors write as if the figs don't exist. Thus, we get all the values shown in the figs, and all the stats shown in the figs. There is no point in this, and all these values should be removed from the Results section; they make it unreadable. Statements such as “30 % lower at 30 °C” should of course be kept. Although there are ideas out there that Results should be shown “by themselves” and not directly commented, I think this is meaningless, given how experiments are planned and conducted. The authors should use the gained space to make primary comments on their results (some of this can be moved in from present discussion); e.g. comment at the end of 3.1.2 on the differences between chow and DIO.

Answer: This is a very good point, thank you. We have removed actual data throughout the result section.

6) Fig 2E should look like 2I and 2M, but just leave the 30 °C value bar empty. The comment on the technical problem should also be in the fig legend.

We agree thus this point and have changed fig. 2E (now fig. 4E) accordingly. Furthermore, we have mentioned the technical problem in the figure legend.

7) line 345: It is a little difficult what the authors want to say with the first statement here – but the “however” in the next sentence seem to imply that in humans (that admittedly do not have fur), adiposity increases insulation. However, when I read the abstract from Nahon it says: “We conclude that larger body sizes possessed reduced LCT as explained by higher BMR related to more lean mass rather than a change in whole-body conductance. Thus, larger individuals with higher lean mass need to be exposed to colder temperatures to activate CIT, not because of increased insulation, but because of a higher basal heat generation. ” The present authors should thus avoid given the impression that mice and humans are different in this respect, according to the data of Nahon.

Thank you for this point. We agree completely and have added the underlined.

“That total fat mass does not correlate strongly with insulation in mice has also been shown by others⁴. The reason for this apparent poor correlation may be influenced by the finding that in mice, fur itself provides about half of the total insulation⁴. Moreover, at least in humans, the insulating factor of fat mass is predominantly mediated by subcutaneous fat mass (thickness)^{25,26}, and since visceral fat constitutes about 80% of total fat mass in rodents²⁷, total fat mass is presumably a suboptimal measure of insulation in rodents. Furthermore, total fat mass may even be a suboptimal measure for insulation in humans since it

has been argued that the increased surface area (and thereby increased heat loss) that inevitably follows with increased fat mass offsets the gain in insulation⁶.

Line 350: what does “same” mean here? And on the fig? I may start to understand – I believe these data are from fig 1 – but what is the °C? For normal calculation of heat loss, I would agree on EE/°C – i.e. the slope of a plot for one mouse where EE was expressed as a function of ambient temperature. I guess then that insulation would be the opposite, °C/EE – but I cannot see the sense in the present representation. If I understand it correctly, the authors should calculate the slope for each mouse and plot that versus fat mass. They could probably plot the data from chow and DIO mice on the same graph. – Concerning the “different” mice, it gets further complicated. As far as I can see, the only choice is to plot the mean EE for all mice as a function of ambient temperature and get a single value for a single mean value of fat mass – and do the same for the DIOs. Thus, all of EG comes out as only two points (!) (similar for FH), and even that is somewhat problematic. OK – this is how I understand it; if the authors have different definitions of insulation etc. the present representations may be adequate but as it is, this fig is not meaningful to me.

We agree that the presentation form of these data was difficult for the reader to assess. Many thanks for your constructive input for improvement of data presentation. The other reviewer has pointed to that our study design for this analysis is sub-optimal since there is a time delay from measurement of body composition and quantification of energy expenditure (9 days). Since neither the normal weight nor the DIO mice remained weight stable throughout the study period, the time difference in data collected may have influenced the outcome of the analysis. We have thus decided to remove all data on insulation.

Fig 4: it is remarkable here that the difference between EE at 30 °C and 22 °C is now more than 100 % (left bar in H versus left bar in B). This should at least be commented.

Answer: The reviewer is right. We have commented this in the discussion with following:

“... At 25 and 30°C, adding a hide and nesting material also significantly reduced EE but responses were quantitatively smaller. At 27.5°C, no differences were observed. (Fig. 6K-N). Notably, EE decreased in these experiments more with increasing temperatures and was in this case ~57% lower at 30°C compared to 22°C”.

Fig 5KL: I do not in general consider it correct to calculate AUC in GTT by the “total” AUC – the initial (fasting) glucose level then dominates the outcome. Here there is no difference in initial glucose level – but the effect of temperature is diminished by this way of calculating. The authors should at least also show a plot of the incremental AUC (the change from initial level).

This point is very well taken, thank you. We have changed the total AUCs to iAUCs and the description of these data. Same differences were compared to the case for total AUCs.

However, the above comments (except for the incomprehensible insulation data) do not take away from the main message of the paper, and with adequate modifications I think this paper will become acceptable for publication.

Many thanks. As you can appreciate, we have taken all your insightful comments into consideration and have acted on all points.

Reviewers' comments:

Reviewer #1 (Remarks to the Author):

The authors have successfully addressed all comments, and there are no additional comments.

Reviewer #2 (Remarks to the Author):

The authors have resubmitted a clearly improved paper. The comments I have are now primarily intended to convince the authors to show the data in such a form that the reader reasonably easily can follow the outcome and so that it becomes more clear what the authors conclude. I think it is a shame if these translationally important data are not communicated in a convincing way and thus may not influence coming studies sufficiently. – I mainly concentrate on the figs.

Figs. 2 and 3. I generally feel that figures should not exaggerate effects. This means that I strongly advocate that y-axes scales start on 0. This means that 2b and 2f should start on 0 – as well as fig 3 b and f. Further, to make issues comparable, the end level on the y-axes should be the same; this means 16 on both 2b and 3b, and 5 on both 2f and 3f, and 1100 on 2j and 3 and 6 on 2h and 3h. – The exception from the 0-rule is of course the RER, but also here the same scale should be used (0.7-1.0).

For figs 4 and 5, the same general point is valid (I won't specify) – but additionally, the subfigs should be at the same place on the fig. The inclusion of fig 5abc totally reshapes fig 5 versus fig 4, to the bewilderment of the reader. 5abc are really not that important and can be moved to a supplement or totally avoided. Thus, 5d-m will be renumbered in parallel with 4a-j. Thereafter the system for showing body weights, fat mass etc. is not paralleled between figs 4 and 5 for no obvious reason; this should be fixed.

In fig 6, the issues with identical scales (for b and f, and d and h) reappear – and in j, RER is suddenly and inadequately presented on an axis that starts at 0 (!).

For figs 7 and 8, the issue with placing the same diagram at the same place on the page reappears. If the authors don't have FGF21 data for fig 8, that space should just be left empty. And by using different scales for the same parameter in the two figs, the vast differences caused by DIO are visually eliminated.

In the Results section, the authors have partly removed the excess of statistical data that is only a duplication of what is documented in the figs. However, only partly removed: all the " $p < 0.05$ " are unnecessary and disturb reading, and statements such as line 165 " $n=7-8$ " again belong to the figs or fig legends, as well as the parenthesis in line 167. We still have unmotivated mentionings of data from the figs, e.g. lines 204-205. The approach in the Results section must be to explain the biological importance of the data in the figs, not to repeat the figs in the text.

The rather long Discussion section would gain by the authors adding concluding statement as subtitles throughout the section, e.g. "Nearly no effect of nesting material" between line 296 and 297, "No indication of insulation by adipose tissue" 207/208, something about blood values 333/334 (my inability to formulate this heading may be caused by the authors not being able to express clearly what were the interesting blood value changes; the section may also be further divided up) and something about BAT 434/435, - and of course "limitations" 450/451 – and "Final conclusion: poor translatability may be due to chronic cold exposure" 465/466.

If the authors want to stress that 30 °C is the "better" temperature, this temperature should be the starting point. Thus, e.g. line 29 should state that "energy expenditure [no reason to enter abbreviations in abstract] increases linearly from 30 to 22 °C and is 50% higher at 22 than at 30 °C in both models". The authors may consider to what degree they would reformulate statements in the rest of the paper to stress that 30 °C in their interpretation is the "better" temperature.

Reposes to reviewers

Reviewer #1 (Remarks to the Author):

The authors have successfully addressed all comments, and there are no additional comments. Dear reviewer. We are pleased that you find that we have addressed your initial comments to satisfaction. Your comments have significantly improved the quality of our manuscript.

Reviewer #2 (Remarks to the Author):

1) The authors have resubmitted a clearly improved paper. The comments I have are now primarily intended to convince the authors to show the data in such a form that the reader reasonably easily can follow the outcome and so that it becomes more clear what the authors conclude. I think it is a shame if these translationally important data are not communicated in a convincing way and thus may not influence coming studies sufficiently. – I mainly concentrate on the figs.

Answer: Dear reviewer. Once again thanks for your constructive and insightful comments and suggestions for improvement. Much appreciated. As specified in the point-to-point responses below, we have acted on all points raised.

2) Figs. 2 and 3. I generally feel that figures should not exaggerate effects. This means that I strongly advocate that y-axes scales start on 0. This means that 2b and 2f should start on 0 – as well as fig 3 b and f. Further, to make issues comparable, the end level on the y-axes should be the same; this means 16 on both 2b and 3b, and 5 on both 2f and 3f, and 1100 on 2j and 3 and 6 on 2h and 3h. – The exception from the 0-rule is of course the RER, but also here the same scale should be used (0.7-1.0).

Answer: Thank you for this comment, which we agree on. We have changed scales on the y-axes to start at 0 and have furthermore standardized the scale between fig. 2 and 3 to the suggested figures. For RER data we have, however, kept the scales as we believe that these data are not meant to be compared between figures since differences in diets (normal weight mice on chow, fig. 2 and DIO mice and 45% HDF, fig. 3) will be an important driver of differences in observed RER values. All other scales have been standardized between figures and now all starts at zero.

3) For figs 4 and 5, the same general point is valid (I won't specify) – but additionally, the subfigs should be at the same place on the fig. The inclusion of fig 5abc totally reshapes fig 5 versus fig 4, to the be wilderment of the reader. 5abc are really not that important and can be moved to a supplement or totally avoided. Thus, 5d-m will be renumbered in parallel with 4a-j. Thereafter the system for showing body weights, fat mass etc. is not paralleled between figs 4 and 5 for no obvious reason; this should be fixed.

Answer: This is a valid point. We have standardized the presentation of data in fig. 4 and 5; both with respect to place of individual subfigures and with respect to Y-axis units. Only exception to this is (for the reasons mentioned in point 2) RER data, which are presented with different units.

4) In fig 6, the issues with identical scales (for b and f, and d and h) reappear – and in j, RER is suddenly and inadequately presented on an axis that starts at 0 (!).

Answer: Thanks for pointing these inconsistencies to our attention. Now fixed.

5) For figs 7 and 8, the issue with placing the same diagram at the same place on the page reappears. If the authors don't have FGF21 data for fig 8, that space should just be left empty. And by using different scales for the same parameter in the two figs, the vast differences caused by DIO are visually eliminated.

Answer: Thank you for this valuable input. We agree and have modified accordingly. Fig. 7 and 8 now have same scales and an empty space have been created in fig. 7 for the missing FGF21 data.

6) In the Results section, the authors have partly removed the excess of statistical data that is only a duplication of what is documented in the figs. However, only partly removed: all the “ $p < 0.05$ ” are unnecessary and disturb reading, and statements such as line 165 “ $n=7-8$ ” again belong to the figs or fig legends, as well as the parenthesis in line 167. We still have unmotivated mentionings of data from the figs, e.g. lines 204-205. The approach in the Results section must be to explain the biological importance of the data in the figs, not to repeat the figs in the text.

Answer: We agree to this advice and have removed sample size information's and p-values from the result section.

7) The rather long Discussion section would gain by the authors adding concluding statement as subtitles throughout the section, e.g. “Nearly no effect of nesting material” between line 296 and 297, “No indication of insulation by adipose tissue” 207/208, something about blood values 333/334 (my inability to formulate this heading may be caused by the authors not being able to express clearly what were the interesting blood value changes; the section may also be further divided up) and something about BAT 434/435, - and of course “limitations” 450/451 – and “Final conclusion: poor translatability may be due to chronic cold exposure” 465/466.

Answer: In fact, we did have subheadings in an earlier version of the manuscript. However, these were removed as we got the impression that the format of the journal did not operate with a structured discussion. However, after having re-examined this, it appears to be an optional choice. We agree that subheadings would ease readability and have thus reintroduced subheadings in the revised discussion.

8) If the authors want to stress that 30 °C is the “better” temperature, this temperature should be the starting point. Thus, e.g. line 29 should state that “energy expenditure [no reason to enter abbreviations in abstract] increases linearly from 30 to 22 °C and is 50% higher at 22 than at 30 °C in both models”. The authors may consider to what degree they would reformulate statements in the rest of the paper to stress that 30 °C in their interpretation is the “better” temperature.

Answer: We agree and have modified the manuscript accordingly.